# An integrated dataset of ground hydrothermal regimes and soil nutrients monitored during 2016-2022 in some previously burned areas in hemiboreal forests in Northeast China

Xiaoying Li [1], Huijun Jin [1, 2, 3] *, Qi Feng [4], Qingbai Wu [1], Hongwei Wang [1, 2], Ruixia He [1],

Dongliang Luo [1], Xiaoli Chang [1, 5], Raul-David Șerban [6], and Tao Zhan [3]

[1] Key Laboratory of Cryospheric Science and Frozen Soil Engineering, Northwest Institute of Eco-Environment and Resources, Chinese Academy of Sciences, Lanzhou 730000, China;

[2] School of Ecology, School of Civil Engineering and Transportation, China-Russia Joint Laboratory of Cold Regions Engineering and Environment, and Permafrost Institute, Northeast Forestry University, Harbin 150040, China;

[3] Ministry of Natural Resources Field Observation and Research Station of Permafrost and Cold Regions Environment in the Da Xing'anling Mountains at Mo'he, Natural Resources Survey Institute of Heilongjiang Province, Harbin 150036, China;

[4] Key Laboratory of Ecohydrology of Inland River Basin, Northwest Institute of Eco-Environment and Resources, Chinese Academy of Sciences, Lanzhou 730000, China;

[5] Hunan University of Science and Technology, Xiangtan, Hunan 411202, China and;

[6] Faculty of Agricultural, Environmental and Food Sciences, Free University of Bozen-Bolzano, Bolzano 39100, Italy

* Corresponding authors: Huijun Jin (hjjin@nefu.edu.cn) at the School of Civil Engineering and Transportation, Northeast Forestry University, Harbin 150040, China

## Abstract:

Under a warming climate, occurrences of wildfires have been increasingly more frequent in boreal forests and arctic tundra during the last few decades. Wildfires can cause radical changes in the forest ecosystems and permafrost environment, such as irreversible degradation of permafrost, succession of boreal forests, rapid and massive

losses of soil carbon stock, and increased periglacial geohazards. Since 2016, we have gradually and more systematically established a network for studying soil nutrients and monitoring the hydrothermal state of the active layer and near-surface permafrost in the northern Da Xing'anling (Hinggan) Mountains in Northeast China. The datasets of soil moisture content (0-9.4 m in depth), soil organic carbon (0-3.6 m), total nitrogen (0-3.6 m), and total phosphorus and potassium (0-3.6 m) were obtained by field sampling and ensuing laboratory tests in 2016. The datasets of ground temperatures (0-20 m) and active layer thickness (2017-2022) were obtained by thermistor cables permanently installed in boreholes or interpolated with these temperatures. The present data can be used to simulate changes in permafrost features under a changing climate and wildfire disturbances and to explore the changing interactive mechanisms of the fire-permafrost-carbon system in the hemiboreal forest. Furthermore, they can provide baseline data for studies and action plans to support the carbon neutralization initiative and assessment of ecological safety and management of the permafrost environment. These datasets can be easily accessed from the National Tibetan Plateau/Third Pole Environment Data Center (https://doi.org/10.11888/Cryos.tpdc.300933, Li and Jin, 2024).

## 1 Introduction

As a key component of the Northern Hemisphere, permafrost and its changes can have substantial consequences for natural and man-made systems (Smith et al., 2022). Moreover, due to its high sensitivity to climate warming, surface disturbances, and human activities, permafrost has undergone extensive degradation during the last six decades (e.g., Biskaborn et al., 2019; Chang et al., 2024; Jin et al., 2000, 2007, 2021, 2022, 2023; Li et al., 2022a; Petrov et al., 2022). As one of the most common natural agents and disturbance factors in boreal forests, wildfires can initiate ecosystem renewal at different spatiotemporal scales (Johnstone et al., 2004; Li et al., 2019). Wildfires impact the permafrost environment first by modifying or altering the ground hydrothermal regimes (Jorgenson et al., 2013; Li et al., 2022b; Yoshikawa et al., 2003), and subsequently by inducing modifications or radical/irreversible changes in

biogeochemical processes (e.g., Fultz et al., 2016; Li et al., 2023; Ping et al., 2010; Xu et al., 2024). In boreal forests, wildfires have become increasingly more frequent in recent decades under a warming climate and increasing human activities (Boyd et al., 2023; Chen et al., 2023; Knorr et al., 2016; Westerling et al., 2006). Moreover, the region immediately south of the Arctic circle (50°N-67°N) experienced a greater number of vegetation fires compared to the Arctic (north of 67°N) in 2001-2020 (Chen et al., 2023). Although the total burned area on Earth may be declining, the fire behavior was worsening in several regions in 2003−2023, particularly the boreal and temperate conifer biome (Cunningham et al., 2024).

In boreal regions, vegetation and soil organic layer are essential buffering and protective layers of the underlying permafrost. The combustion of all vegetation cover and partial or complete removal of the insulating organic layer have direct hydrothermal impacts on permafrost. It reduces the land surface albedo, increases ground surface and cryosol/ice exposure to direct solar radiation, and weakens the cooling effects of vegetative shading and evapotranspiration (Johnstone et al., 2010; Nossov et al., 2013; Shur and Jorgenson, 2007; Yoshikawa et al., 2003). All of these contribute to higher ground surface temperature and more heat transferred into the ground, resulting in a rapid ground warming and sharp deepening of the active layer (Li et al., 2022b; Michaelides et al., 2019; Nossov et al., 2013; Smith et al., 2015). In Interior Alaska, organic layer thickness decreased from 21 to 4 cm after fire, resulting in thaw depth increasing from 72 to 152 cm, mean annual surface temperature rising from −0.6 to +2.1°C and mean annual deep temperature going up from −1.7 to +0.4°C (Nossov et al., 2013). In the boreal zone, 6-11 years after fire, mean annual ground temperature (MAGT) increased by 1.5-2.3°C (Li et al., 2021; Munkhjargal et al., 2020; Nossov et al., 2013; Smith et al., 2015), even mean annual ground surface temperatures in burned areas were still 2-3°C higher than those in unburned areas 80 years after fire (Brown et al., 2015). Meanwhile, 25 years after fire, the active layer thickness (ALT) could increase by 2.75 m, from the initial value of 45 cm, and ALT could not recover to the pre-fire level even 36 years after fire (Viereck et al., 2008). In Central Siberia, it

generally takes 70-80 years for the active layer to return to the pre-fire state (Kirdyanov et al., 2020). Forest fires also can cause significant changes in soil moisture contents, which in turn affects ground thermal regimes (Nossov et al., 2013). Due to the fire-induced thaw of permafrost, the charred moss layers with lowered infiltration rates, lower transpiration rate and reduced evapotranspiration in severely burned areas, surface soil moisture contents (generally less than 30 cm in depth) at burned sites were significantly higher than those at unburned sites (Kopp et al., 2014; Potter and Hugny, 2020; Yoshikawa et al., 2003). However, affected by soil texture, permafrost thaw after fire can also lead to a decrease in soil moisture contents (Li et al., 2022b; Nossov et al., 2013). In summary, in a short term, forest fires will decrease rates of transpiration, raising soil moisture contents; in a long-term (more than a decade), the increased ALT and recovery of vegetation will reduce soil moisture content at burned sites as compared to that at unburned sites (Yoshikawa et al., 2003). Moreover, changes in ground hydrothermal regimes and ALT would decline and progressively dwindle with ecosystem recovery and organic layer regrowth over time under a stable or cooling climate (e.g. Holloway et al., 2020; Rocha et al., 2012).

Arctic-boreal permafrost soils contain between 1100-1500 Pg (1 Pg=$10^{15}$ g) carbon, approximately twice of the carbon pool in the atmosphere (Hugelius et al., 2014), and accounting for nearly half of the global belowground organic carbon pool (O'Donnell et al., 2011a). Wildfire disturbances have important and long-term ramifications for terrestrial carbon cycling and carbon stocks (Chen et al., 2022; Dieleman et al., 2022; Genet et al., 2013; O'Donnell et al., 2011a, 2011b). Unlike gradual thawing, abrupt changes after fires in ground hydrothermal regimes often disrupt the entire soil profile and initiate or aggravate carbon loss from deep permafrost soils (Jones et al., 2015; Turetsky et al., 2019). Therefore, the combustion of vegetation and the subsequent thaw of permafrost have resulted in rapid releases of large amounts of carbon and nitrogen into the atmosphere as greenhouse gases (Mack et al., 2011, 2021; Taş et al., 2014). Furthermore, over a short time, abrupt permafrost thaw would possibly result in emitting more methane than gradual thaw (Koven et al., 2015).

Therefore, in the boreal permafrost region, wildfire exacerbate rates of permafrost thaw and alter soil organic carbon dynamics in both organic and mineral soils. In addition to soil organic carbon, forest fires potentially also reduce soil nitrogen contents, inducing shifts in nutrient cycling in the boreal forest and permafrost regions (Certini, 2005; Knicker, 2007; Kolka et al., 2017). However, there are inconsistent reports on the effects of forest fire on soil phosphorus and potassium. Some studies show a significant post-fire reduction in phosphorus and potassium while other studies indicate an evident increase after light burns, but a reduction after severe burns, and nearly unchanged stocks of potassium and phosphorus (Gu et al., 2010; Neff et al., 2005; Zhao et al., 1994). As a result, wildfires in boreal permafrost regions had been considered to trigger strong positive feedbacks on climate warming *via* massive emissions of biogenic major greenhouse gases (Koven et al., 2015; Ramm et al., 2023).

Located on the southern margin of Eastern Asian boreal forests and permafrost regions, the Da Xing'anling (Hinggan) Mountains in Northeast China are prone to frequent and massive wildfires. The Xing'an permafrost here is controlled or strongly affected by many local factors, such as dense vegetation cover, thick organic layer, stable snow cover, and anthropic development (Jin et al., 2007; Șerban et al., 2021; Wang et al., 2024). The warm and thin permafrost in the Da Xing'anling Mountains in Northeast China is located in the discontinuous permafrost zone. Therefore, this ecosystem-dominated (driven, modified, or protected) permafrost is sensitive to climate warming and wildfires (Shur and Jorgenson, 2007). Compared with the Arctic permafrost region, the permafrost monitoring network in this region has been established only recently, with inadequately readily accessible and shared permafrost data. Similarly, the permafrost monitoring data in the burned areas in the boreal permafrost region in China are meagre in comparison with those other northern countries or regions, but they are increasing. Prior to the early 1980s, there was little research on wildfire impacts on the permafrost environment in Northeast China. There were only a few occasional fire-related geocryological studies in the early 1990s and limited site-specific measurements of soil temperature and moisture content in the

active layer and near-surface (≤20 m in depth) permafrost near the Amu'er town, northern Heilongjiang Province (Liang et al., 1991; Zhou et al., 1993). Moreover, research on fire impacts on soil carbon and nitrogen pools and cycles in the Xing'an permafrost in Northeast China has just started and is still at its fledgling stage. Due to the cold and arid climate in winter and spring, complex mountain topography, and dense hemiboreal vegetation in the region, fire regimes are often complex. In addition, burned areas are often located in pristine forest areas far away from roads, making it challenging to timely and/or readily access and study fire impacts. Therefore, it is difficult to systematically understand and quantitatively evaluate the effects of wildfires on ground hydrothermal regimes and carbon stocks at different spatiotemporal scales (Li et al., 2021).

To address the abovementioned issues, since 2016, an observation system has been gradually established for ground hydrothermal regimes and soil nutrient contents in the northern Da Xing'anling Mountains. This dataset can provide important supportive data for studying permafrost landscapes, carbon stocks, and boreal ecology and hydrology. It can also provide important references for the management of land and water resources and ecological environment after wildfire disturbances in Northeast China, particularly in forested hemiboreal permafrost regions. In Section 2 of this paper, we first introduce the comprehensive observation network of permafrost and soil nutrients in the northern Da Xing'anling Mountains. The design of the monitoring network of ground hydrothermal regimes and systematic observations of soil nutrient contents, and evaluation of data quality are given in Section 2. In Section 3, observations of permafrost hydrothermal regimes and soil nutrients that provide a 6-year-long dataset are described and briefly interpreted with a focus on major features of the observation network for better understanding of the dataset structure and contents. The data availability and accessibility are provided in Section 4, and; in Section 5, major conclusions and prospects are given. This dataset provides important input for the model simulations of permafrost changes under fire disturbances and a warming climate, especially those rapid and abrupt degradation of the Xing'an permafrost and resultant

periglacial phenomena, such as thermokarst, thaw settlement, and ground surface subsidence and ponding. It is useful for analyzing the interactive hydrothermal and cyclic mechanisms of the wildfires-permafrost-carbon system in the hemiboreal forest.

## 2 Monitoring networks and data processing

### 2.1 Study area descriptions and monitoring networks

A permafrost monitoring network has been established in four burned areas in the northern Da Xing'anling Mountains in Northeast China in boreal forest and discontinuous permafrost regions (Figure 1). Two are located in shrub wetlands in Mo'he city (MH) and Gulian town (GL) in northern Heilongjiang Province. The other two are located in larch forests in Alongshan (AL) and Mangui (MG) towns in the northeastern part of Inner Mongolia. The network includes eight sites in the four burned areas with two fire severity (severely burned (S) and unburned (U)) from 1987 to 2015 (the fire severity division method was shown in "2.2 *Fire severity*" section). The studied forest fire in MH (with severely burned (MH-S) and unburned (MH-U) sites) occurred on 6 May 1987, with a burned-over area of $1.01 \times 10^6$ ha; that in GL (with severely burned (GL-S)) and unburned (GL-U) sites), on 28 July 2002, 1,121 ha; AL (with severely burned (AL-S) and unburned (AL-U) sites), on 10 May 2009, 930 ha, and; MG (with severely burned (MG-S) and unburned (MG-U) sites), on 12 July 2015, 237 ha.

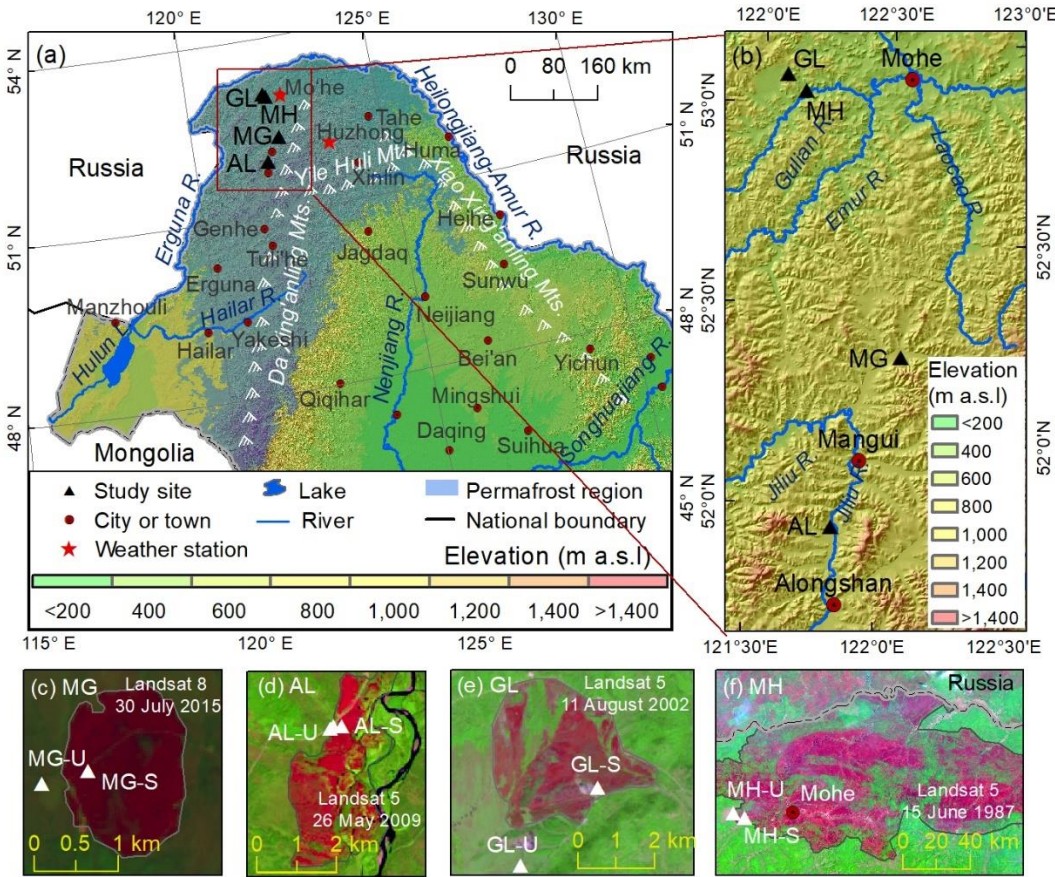


Figure 1. Location of the study areas and sites in the northern Da Xing'anling Mountains, Northeast
China.
Notes: The base map of permafrost distribution is modified from Li et al. (2022c). The light blue
areas in Figure 1a are the permafrost region. Figures 1c to 1f are the false-color composite image of
the remote sensing image; the burned areas are marked as pink, and the unburned areas are marked
as green.
The study areas are characterized by a cold temperate continental climate. In the
study areas of GL and MH, based on the data of nearby Mo'he weather station from
1960 to 2020, mean annual air temperature (MAAT) ranged from –6.2 to –2.4°C, with
an average rate of climate warming at 0.3°C per decade; annual precipitation was 274-
675 mm, with a slight average wetting trend of 13.8 mm per decade. In the study areas
of MG and AL, based on the data of nearby Huzhong weather station from 1974 to 2020,
MAAT varied from –5.2 to –2.0°C, with the same climate warming rate as that of
Mo'he (0.3°C/decade); annual precipitation was 272-749 mm, showing an appreciable
average wetting rate of 3.1 mm per decade. Precipitation fell concentratively in the form

of rain from June to August, accounting for 62%-65% of the annual total. Snow cover generally lasted from October to the next May, with maximum snow depths at 40-50 cm.

The four study areas were selected to observe post-fire changes in permafrost features and soil nutrient conditions (Table 1). This monitoring network includes eight boreholes and soil profiles, and major elements of the observational network for ground temperature, ALT, soil moisture content (SMC), soil organic carbon (SOC), total nitrogen (TN), total phosphorus (TP), and total potassium (TK). The MAGT at the depth of zero annual amplitude ($D_{ZAA}$, generally at 10-15 m in depth) ranged from $-3.25$ to $-0.56°C$, and measured ALT varied from 1.0 to 3.75 m. The four study areas were all found in the zones of discontinuous permafrost, with poor drainage in lowlands and intermontane basins or valleys. The soils in the study area are mainly Histosol and Gelisols (Soil Survey Staff, 2014). Before fires, vegetation was dominated by the Xing'an larch (*Larix gmelinii*) forest, generally with an understory mainly consisting of the shrubs *Ledum palustre* and *Vaccinium uliginosum*, with an organic layer of 55-60 cm in thickness. After fires, the vegetation of burned over areas became gradually dominated by white birch (*Betula platyphylla*) and dwarf bog birch (*Betula fruticosa* Pallas), with an organic layer of 20-30 cm in thickness. At severe burned sites in AL, GL, and MH, measurements of organic matter thickness were taken 7, 14, and 29 years after fires, so it was possible that the organic layer thickness exceeded 20 cm due to the re-accumulation of organic matter. At severe burned site in MG, the organic matter residue after combustion was in a fluffy state with the thickness of 20 cm. When the re-accumulation or residual organic matter exceeded 20 cm, this would slow the rate of active layer thickening and soil temperature increase after fires, as well as the permafrost would gradually recover with the re-accumulation of organic layer.

Table 1. Characteristics of the eight study sites for monitoring the thermal state and soil nutrients of the active layer and near-surface permafrost in the northern Da Xing'anling Mountains in Northeast China

| Study areas and sites | Lat | Long. | Elev. | Veget | Organic | Drainage | Fire |
| --- | --- | --- | --- | --- | --- | --- | --- |

| | | (ºN) | (ºE) | (m a. s. l.) | -ation | layer thickness (cm) | | severity |
|---|---|---|---|---|---|---|---|---|
| MG | MG-S | 52.2765 | 122.2891 | 710 | Larch forest | 20 | Somewhat poor | Severely burned |
| (Mangui) | MG-U | | | | | 55 | Poor | Unburned |
| AL | AL-S | 51.8868 | 121.9067 | 669 | Larch forest | 25 | Moderately good | Severely burned |
| (Alongshan) | AL-U | | | | | 55 | Poor | Unburned |
| GL | GL-S | 53.0432 | 122.0504 | 582 | Shrub wetland | 30 | Somewhat poor | Severely burned |
| (Gulian) | GL-U | | | | | 60 | Poor | Unburned |
| MH | MH-S | 52.9859 | 122.1115 | 486 | Shrub wetland | 30 | Somewhat poor | Severely burned |
| (Mo'he) | MH-U | | | | | 60 | Poor | Unburned |

The horizontal distance between MG-U and MG-S was about 200 m, with the MG-
U on the edge of the burned area. Observations of ground temperatures began in
February 2017 (two years after fire). At MG-U in the Xing'an larch (*Larix gmelinii*)
dominated forest, all larch trees at MG-S were burned to death, and low shrubs and
herbs were found in 2022. The horizontal distance between AL-U and AL-S was less
than 100 m, with the AL-U on the edge of the burned area. Observations of ground
temperatures began in February 2017 (eight years after fire). The vegetation was the
Xing'an larch forest at AL-U, and; it was the broad-leaved forest (birch) at AL-S. We
selected GL-S and GL-U sites about 2 km apart from each other. Measurements of
ground temperatures began in February 2017 (15 years after fire). The vegetation was
the shrub wetland at GL-U and GL-S. MH-S and MH-U sites were about 5 km apart.
Observations of ground temperatures began in February 2017 (30 years after fire). The
ecosystem was characteristic of shrub wetlands at MH-U and MH-S.

## 2.2 Fire severity

Normalized Burn Ratio (NBR) and differential Normalized Burn Ratio (dNBR)
are often used to assess the forest fire severity (Cocke et al., 2005; Li et al., 2022b), and

the calculation formulas are as follows:

$$NBR = (\rho_{NIR} - \rho_{MIR})/(\rho_{NIR} + \rho_{MIR}) \qquad (1)$$

$$dNBR = NBR_{prefire} - NBR_{postfire} \qquad (2)$$

where $\rho_{NIR}$ and $\rho_{MIR}$ are the reflectivity values of pixel from the near-infrared (NIR) and middle-infrared (MIR) bands, and; $NBR_{prefire}$ and $NBR_{postfire}$ are the values of NBR before and after fire.

According to the Cocke et al. (2005) and Roy et al. (2006), the *dNBR* values of 0.241 and 0.57 are the critical values for the division between lightly and moderately burned, and moderately and severely burned. Therefore, through vegetation burn status and the comparison with *dNBR* values (Key and Benson, 2006; Escuin et al., 2008), fire severity is thus divided into four categories: severely burned (dNBR $\geqslant$ 0.571), moderately burned (0.241-0.570), lightly burned (0.051-0.240), and unburned ($\leqslant$0.050) (Cocke et al., 2005). In the lightly and moderately burned areas, there were difficulties in drilling and/or monitoring due to device malfunction or damage. In addition, the permafrost environment changes more significantly after severe burns. Therefore, only sites of two levels of fire severity (severely burned and unburned) were chosen for the abovementioned four areas (Mangui/MG, Alongshan/AL, Gulian/GL and Mo'he/MH) to study post-fire changes in ground hydrothermal regimes and soil nutrients.

## 2.3 Site instrumentation and laboratory analysis

At each site of unburned and severe burned, a 20-m-deep borehole was drilled and instrumented in October 2016 to monitor ground temperatures (eight boreholes in total) (Figure 2). Ground temperatures were monitored with 0.5-m depth intervals at depths of 0–5 m and then with 1-m depth intervals at depths of 5-20 m by thermistor cables permanently installed in boreholes and manually measured from February 2017. All thermistors were assembled and calibrated at the Key Laboratory of Cryospheric Science and Frozen Soil Engineering, Northwest Institute of Eco-Environment and Resources (renamed from the merger of the former State Key Laboratory of Frozen Soil Engineering and the State Key Laboratory of Cryosphere Science, Cold and Arid

Regions Environmental and Engineering Research Institute), Chinese Academy of Sciences. Since February 2017, ground temperatures at these boreholes were manually measured thrice monthly (Table 2), or occasionally once or twice monthly due to traffic difficulty or control, by a multi-meter Fluke 189® device. According to the measured soil temperatures during the observation period, the isotherms of soil temperature in the vertical profile at depths of 0-20 m were drawn, and then the 0°C isotherms were delineated for each borehole. The values of ALT were then determined, using linear extrapolation of seasonally and progressively changing ground temperature distribution with depth, for each borehole and each year according to the deepest position of the $0°C$ isotherms in the year.

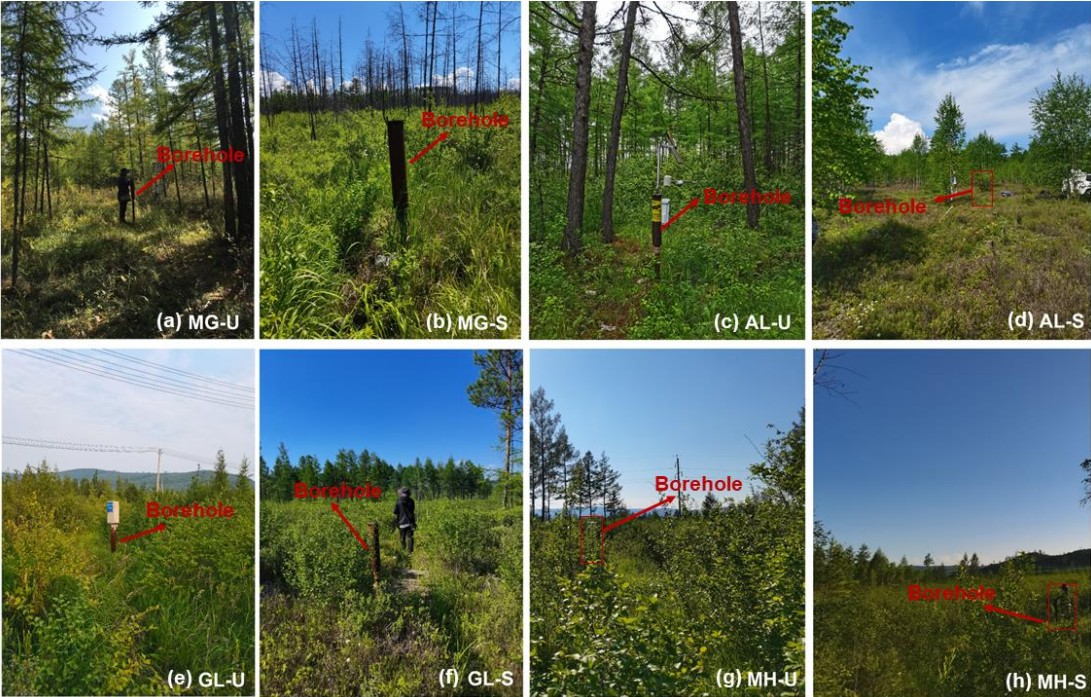

Figure 2. Photos of the study sites with different vegetation cover and the position of the 20 m deep boreholes for monitoring the ground temperature in the northern Da Xing'anling Mountains in Northeast China in 3-5 July 2022.

Notes: Figures 2a and 2b were the borehole for observation of ground temperature at Xing'an larch forest severe burned and light burned sites in MG; Figures 2c and 2d were the borehole for observation of ground temperature in a Xing'an larch forest at severe burned and light burned sites in AL; Figures 2e and 2f were the borehole for observation of ground temperature in shrub wetlands at severe burned and light burned sites in GL; Figures 2g and 2h were the borehole for observation of ground temperature at shrub wetlands severe burned and light burned sites in MH.

Table 2. Monitoring data for the eight sites of soil nutrients and ground temperature boreholes for studying fire impacts on the permafrost environment in the northern
Da Xing'anling Mountains in Northeast China

| Study sites | Monitoring depths (m) | | | Time period | Monitoring frequency |
| | Soil nutrients | Soil gravimetric moisture content (SMC) | Ground temperature | | |
| --- | --- | --- | --- | --- | --- |
| MG-U | 0.1, 0.2, 0.3, 0.4, 0.5, 0.6, 0.7, 0.8, 0.9, 1.0, 1.1, 1.2, 1.3, 1.4, 1.5, 1.6, 1.7, 1.8, 1.9, 2.0, 2.1, 2.2, 2.3, 2.4, 2.5 | 0.2, 0.3, 0.4, 0.5, 0.6, 0.7, 0.8, 0.9, 1.0, 1.1, 1.2, 1.3, 1.4, 1.5, 1.6, 1.7, 2.0, 2.5, 2.7 | 0.0, 0.2, 0.5, 1.0, 1.5, 2.0, 2.5, 3.0, 3.5, 4.0, 5.0, 6.0, 7.0, 8.0, 9.0, 10.0, 11, 12, 13, 14, 15, 16, 17, 18, 19, 20 | 2016; 2016; 2017-2022 | Once; Once; Thrice/ month |
| MG-S | 0.1, 0.2, 0.3, 0.4, 0.5, 0.6, 0.7, 0.8, 0.9, 1.0, 1.1, 1.2, 1.3, 1.4, 1.5, 1.6, 1.7, 1.8, 1.9, 2.0, 2.1, 2.2, 2.3, 2.4, 2.5, 2.6 | 0.2, 0.3, 0.4, 0.5, 0.6, 0.7, 0.8, 0.9, 1.0, 1.1, 1.2, 1.3, 1.4, 1.5, 1.6, 1.7, 1.8, 1.9, 2.0, 2.1, 2.2, 2.6, 4.6, 5.6, 6.1, 7.6 | 0.0, 0.2, 0.5, 1.0, 1.5, 2.0, 2.5, 3.0, 3.5, 4.0, 5.0, 6.0, 7.0, 8.0, 9.0, 10.0, 11, 12, 13, 14, 15, 16, 17, 18, 19, 20 | 2016; 2016; 2017-2022 | Once; Once; Thrice/ month |
| AL-U | 0.1, 0.2, 0.3, 0.4, 0.5, 0.6, 0.7, 0.8, 0.9, 1.0, 1.1, 1.2, 1.3, 1.4, 1.5, 1.6, 1.7, 1.8, 1.9, 2.0, 2.1, 2.2, 2.3, 2.4, 2.5, 2.6, 2.7, 2.8, 2.9, 3.0 | 0.1, 0.2, 0.3, 0.4, 0.5, 0.6, 0.7, 0.8, 0.9, 1.0, 1.1, 1.2, 1.3, 1.4, 1.5, 1.6, 1.7, 1.8, 1.9, 2.0, 2.1, 2.2, 2.3, 2.4, 2.5, 2.6, 2.7, 2.8, 2.9, 3.0, 3.1, 3.2, 3.5, 4.0, 4.5, 5.0, 5.5, 5.9, 6.4, 9.4 | 0.0, 0.2, 0.5, 1.0, 1.5, 2.0, 2.5, 3.0, 3.5, 4.0, 5.0, 6.0, 7.0, 8.0, 9.0, 10.0, 11, 12, 13, 14, 15, 16, 17, 18, 19, 20 | 2016; 2016; 2017-2022 | Once; Once; Thrice/ month |
| AL-S | 0.1, 0.2, 0.3, 0.4, 0.5, 0.6, 0.7, 0.8, 0.9, 1.0, 1.1, 1.2, 1.3, 1.4, 2.1, 2.2, 2.3, 2.4, 2.5, 2.6, 2.7, 2.8 | 0.2, 0.3, 0.4, 0.5, 0.6, 0.7, 0.8, 0.9, 1.1, 1.4, 1.5, 1.7, 2.0, 2.2, 2.4, 2.6, 2.8, 2.9, 3.1, 3.4, 3.6, 4.0, 4.1, 4.5, 4.8, 5.5, 6.0, 7.0, 7.5 | 0.0, 0.2, 0.5, 1.0, 1.5, 2.0, 2.5, 3.0, 3.5, 4.0, 5.0, 6.0, 7.0, 8.0, 9.0, 10.0, 11, 12, 13, 14, 15, 16, 17, 18, 19, 20 | 2016; 2016; 2017-2022 | Once; Once; Thrice/ month |
| GL-U | 0.1, 0.2, 0.3, 0.4, 0.5, 0.6, 0.7, 0.8, 0.9, 1.0, 1.1, 1.4, 1.5, 1.6, 1.7, 1.8, 1.9, 2.0, 2.1, 2.2, 2.3, 2.4, 2.5, 2.6, 2.7, 2.8, 2.9, 3.0, 3.1, 3.4, 3.5, 3.6 | 0.1, 0.2, 0.3, 0.4, 0.5, 0.6, 0.7, 0.8, 0.9, 1.0, 1.1, 1.3, 1.4, 1.5, 1.6, 1.7, 1.8, 1.9, 2.0, 2.7, 2.8, 2.9, 3.0, 3.1 | 0.0, 0.2, 0.5, 1.0, 1.5, 2.0, 2.5, 3.0, 3.5, 4.0, 5.0, 6.0, 7.0, 8.0, 9.0, 10.0, 11, 12, 13, 14, 15, 16, 17, 18, 19, 20 | 2016; 2016; 2017-2022 | Once; Once; Thrice/ month |
| GL-S | 0.1, 0.2, 0.3, 0.4, 0.5, 0.6, 0.7, 0.8, 0.9, 1.0, 1.2, 1.3, 1.4, 1.5, 2.0, 2.1, 2.2, 2.4, 2.5, 2.6, 2.7, 2.8 | 0.1, 0.2, 0.3, 0.8, 2.0, 2.4, 2.7, 3.6, 4.2, 4.7, 5.6, 8.4 | 0.0, 0.2, 0.5, 1.0, 1.5, 2.0, 2.5, 3.0, 3.5, 4.0, 5.0, 6.0, 7.0, 8.0, 9.0, 10.0, 11, 12, 13, 14, 15, 16, 17, 18, 19, 20 | 2016; 2016; 2017-2022 | Once; Once; Thrice/ month |
| MH-U | 0.1, 0.2, 0.3, 0.4, 0.5, 0.6, 0.7, 0.8, 0.9, 1.0, 1.1, 1.4, 1.5, 1.6, 1.7, 1.8, 1.9, 2.0, 2.1, 2.2, 2.3, 2.4, 2.5, 2.6, 2.7, 2.8, 2.9, 3.0, 3.1, 3.4, 3.5, 3.6 | 0.1, 0.2, 0.3, 0.4, 0.5, 0.6, 0.7, 0.8, 0.9, 1.0, 1.1, 1.3, 1.4, 1.5, 1.6, 1.7, 1.8, 1.9, 2.0, 2.7, 2.8, 2.9, 3.0, 3.1 | 0.0, 0.2, 0.5, 1.0, 1.5, 2.0, 2.5, 3.0, 3.5, 4.0, 5.0, 6.0, 7.0, 8.0, 9.0, 10.0, 11, 12, 13, 14, 15, 16, 17, 18, 19, 20 | 2016; 2016; 2017-2022 | Once; Once; Thrice/ month |
| MH-S | 0.1, 0.2, 0.3, 0.4, 0.5, 0.6, 0.7, 0.8, 0.9, 1.0, 1.1, 1.2, 1.3, 1.4, 1.5, 1.6, 1.7, 1.8, 1.9, 2.0 | 0.1, 0.2, 0.3, 0.4, 0.5, 0.6, 0.7, 0.8, 0.9, 1.0, 1.1, 1.2, 1.3, 1.4, 1.5, 1.6, 1.7, 1.8, 1.9, 2.0, 2.3, 3.6 | 0.0, 0.2, 0.5, 1.0, 1.5, 2.0, 2.5, 3.0, 3.5, 4.0, 5.0, 6.0, 7.0, 8.0, 9.0, 10.0, 11, 12, 13, 14, 15, 16, 17, 18, 19, 20 | 2016; 2016; 2017-2022 | Once; Once; Thrice/ month |

Notes: Soil nutrients and SMC were observed once in 2016, and soil temperatures were observed thrice monthly in 2017-2022.
While drilling in 2016, soil samples were collected from depths of 0-9.4 m at
intervals of 0.1-3.0 m, with a total of 402 soil samples. Three replicas were collected at
the same depth and then three samples were evenly mixed into one. At depths of 0-3.0
m, samples were collected every 10 cm in depth in soil strata with more significant
changes of soil organic matter and lithology near the ground surface. At depths of 3.0-
9.4 m, samples were collected based on lithological similarity or changes in soil or rock
strata, rather than at an equal depth interval of 0.1 m. Therefore, at depths of 0-3 m,
there were generally a set of data at a regular depth interval of 10 cm, but at depths of
3-10 m, the depth intervals of datasets varied substantially. One part of the soil samples
was collected using a cutting ring and stored in an 100-cm$^3$ aluminum specimen box
and immediately weighed (soil wet weight). Then, the samples were transported to the
laboratory and dried at 105°C to obtain soil dry weight. Finally, gravimetrically-based
SMC was calculated by the mass of soil before and after drying. The other part of the
soil samples was collected and stored in zip-lock bags and timely brought back to the
laboratory for air-drying, then passed through a 2-mm sieve for chemical analysis. SOC
and TN contents were measured by potassium dichromate oxidation reduction and
Kjeldahl nitrate boiling fluid injection methods, respectively (Nelson et al., 1982). TP
and TK contents were determined by the methods of Mo-Sb colorimetry and flame
photometry, respectively (Sun et al., 2011). These data are shown as mean ± standard
error (SE). Changes in ground temperatures and soil chemical properties were analyzed
using the space-for-time chronosequence approach (Mack et al., 2021).
**2.4 Data quality check**

The measurement accuracy of ground temperature was ±0.05°C in the range of

−30 to +30°C, but ± 0.1°C in those of −45 to −30°C and +30 to +50°C. From 2020 to
2022, due to the breakout and persistence of the COVID-19 pandemic, some data were
not timely collected, affecting the sampling intervals. Ground temperature data were
collected manually thrice monthly since February 2017, and after the outbreak of the
COVID-19 pandemic, the data were recorded once or twice monthly. In addition, some
data were missing because of damaged, broken, or destroyed probes, solar panel

batteries, or dataloggers. From 6 February 2017 to 22 November 2022, a total of 28,890 data records were collected, of which 178 NA (not available) data were resulted from probe damage, thus 28,712 valid data were collected. All the missing data were near the ground surface, at a soil layer at depths between 0 and 5 cm. At MG-U, AL-U, AL-S, GL-S, and MH-S, all data were available. Of the 178 NA data, 74 were at MG-S (from 17 September 2019 to 22 November 2022), 52 at GL-U (from 20 July 2019 to 13 February 2022), and 52 at MH-U (from 20 July 2019 to 13 February 2022) sites. Data of soil temperatures from manually monitored boreholes were quality-controlled for each measurement. Some studies have also shown that this method of monitoring ground temperature using drilling and probes is one of the most accurate, reliable, and intuitive methods for long-term monitoring of permafrost data (Chang et al., 2022; Li et al., 2022a, 2024; Zhao et al., 2021). Before the analysis of soil nutrient data and SMC data, we conducted outlier tests to ensure the accuracy of the data. These tests showed that all the data have no outliers and the samples are representative. There was a total of 840 soil nutrient data and 195 SMC data.

## 3 Data descriptions and evaluation

### 3.1 Changes in ground temperatures of near-surface permafrost

Ground temperatures at depths of 0-20 m in the active layer and near-surface permafrost showed remarkable seasonal dynamics (Figures 3 and 4). The amplitudes of changes in ground temperature decreased exponentially with increasing depth. At depths of 0-1 m, changes in MAGT at eight sites were larger 1.5-10.2°C than those at 1-20 m (Figures 3a to 3d).

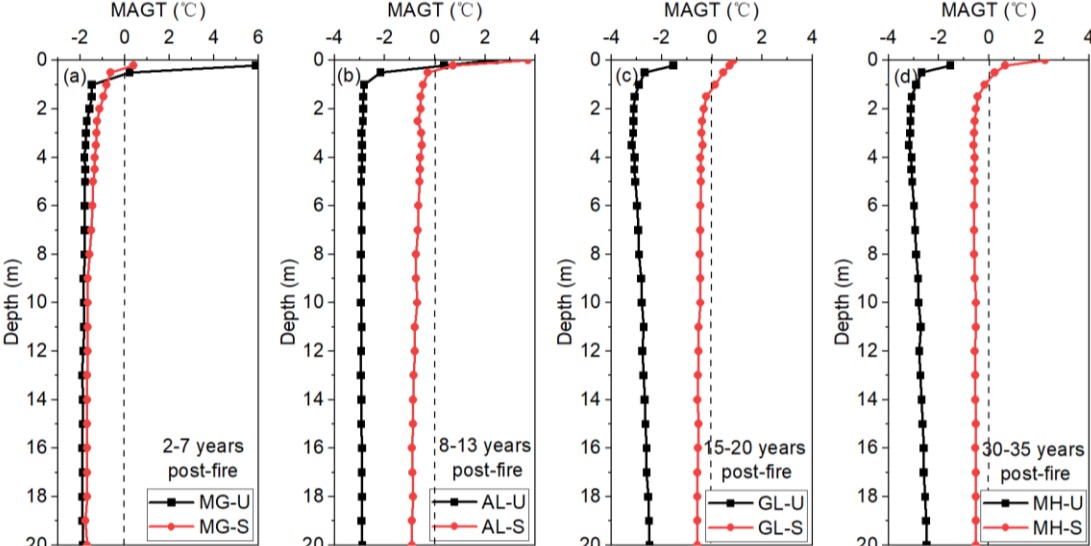

Figure 3. Mean annual ground temperatures (MAGTs) from 2017 to 2022 at the unburned and severely burned sites in the four areas on the western flank of the northern Da Xing'anling Mountains in Northeast China

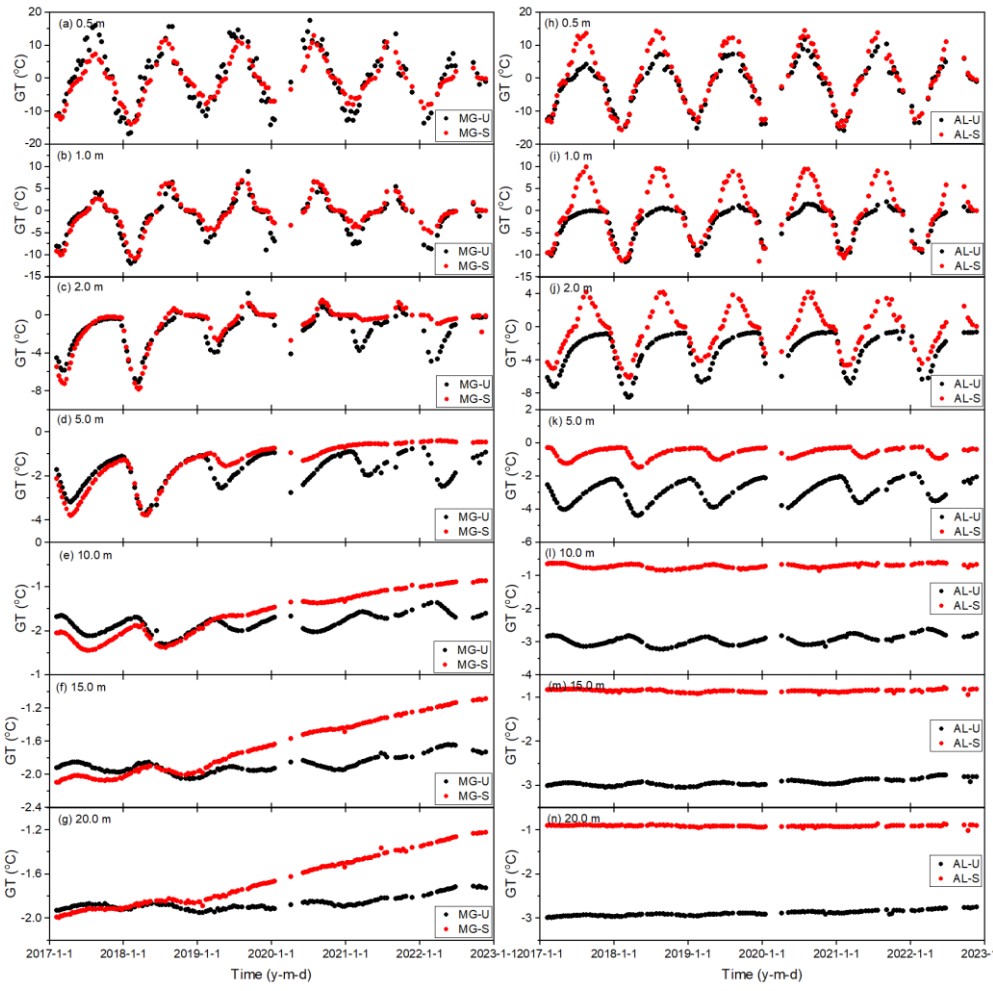

Figure 4. Variability of ground temperatures at depths of 0–20 m at Xing'an larch forest sites in Mangui (MG) and Alongshan (AL) on the western flank of the northern Da Xing'anling Mountains in Northeast China during the period from 2017 to 2022.

MAGTs lowered with increasing depths, the temperature difference between 0.5
and 20 m in depth was 0.2-2.1℃ (Table 3). From 2017 to 2022, ground temperature
fluctuated in a sinusoidal pattern at depths of 0.5 to 2.0 m, and this dynamic change
gradually disappeared with increasing depth (Figures 3a to 3g and 5a to 5g). At the
depth of 5 m, ground temperature was subzero or perennially cryotic (Figures 4d, 4k,
5d, and 5k). At eight sites, from 2017 to 2022, ground temperatures showed an
increasing trend of 0.01-0.69℃/yr at depths of 0.5-20 m. The increase rate was the
largest at AL-U (0.03-0.69℃/yr), and; the lowest, at AL-S and GL-S (all were 0.01-
0.37℃/yr) (Figures 4a to 4g and Figures 5a to 5g).
Table 3. Mean annual ground temperatures (MAGTs) at each of the seven measured depths at
unburned and severely burned sites in the four areas on the western flank of the northern Da
Xing'anling Mountains in Northeast China during the period from 2017 to 2022

| Depth (m) | 0.5 | | 1.0 | | 2.0 | | 5.0 | | 10 | | 15 | | 20 | |
|---|---|---|---|---|---|---|---|---|---|---|---|---|---|---|
| Fire severity | U | S | U | S | U | S | U | S | U | S | U | S | U | S |
| MG | 0.2 | -0.6 | -1.5 | -0.8 | -1.6 | -1.1 | -1.7 | -1.4 | -1.8 | -1.6 | -1.9 | -1.7 | -1.9 | -1.7 |
| AL | -2.2 | -0.3 | -2.8 | -0.5 | -2.9 | -0.6 | -2.9 | -0.6 | -2.9 | -0.7 | -2.9 | -0.9 | -2.9 | -0.9 |
| GL | -2.7 | 0.5 | -2.9 | 0.1 | -3.1 | -0.3 | -3.1 | -0.4 | -2.8 | -0.5 | -2.6 | -0.5 | -2.5 | -0.6 |
| MH | -2.7 | 0.2 | -2.9 | -0.2 | -3.1 | -0.5 | -3.1 | -0.6 | -2.8 | -0.5 | -2.6 | -0.5 | -2.5 | -0.5 |

Notes: U stands for unburned sites, and; S, severely burned sites.

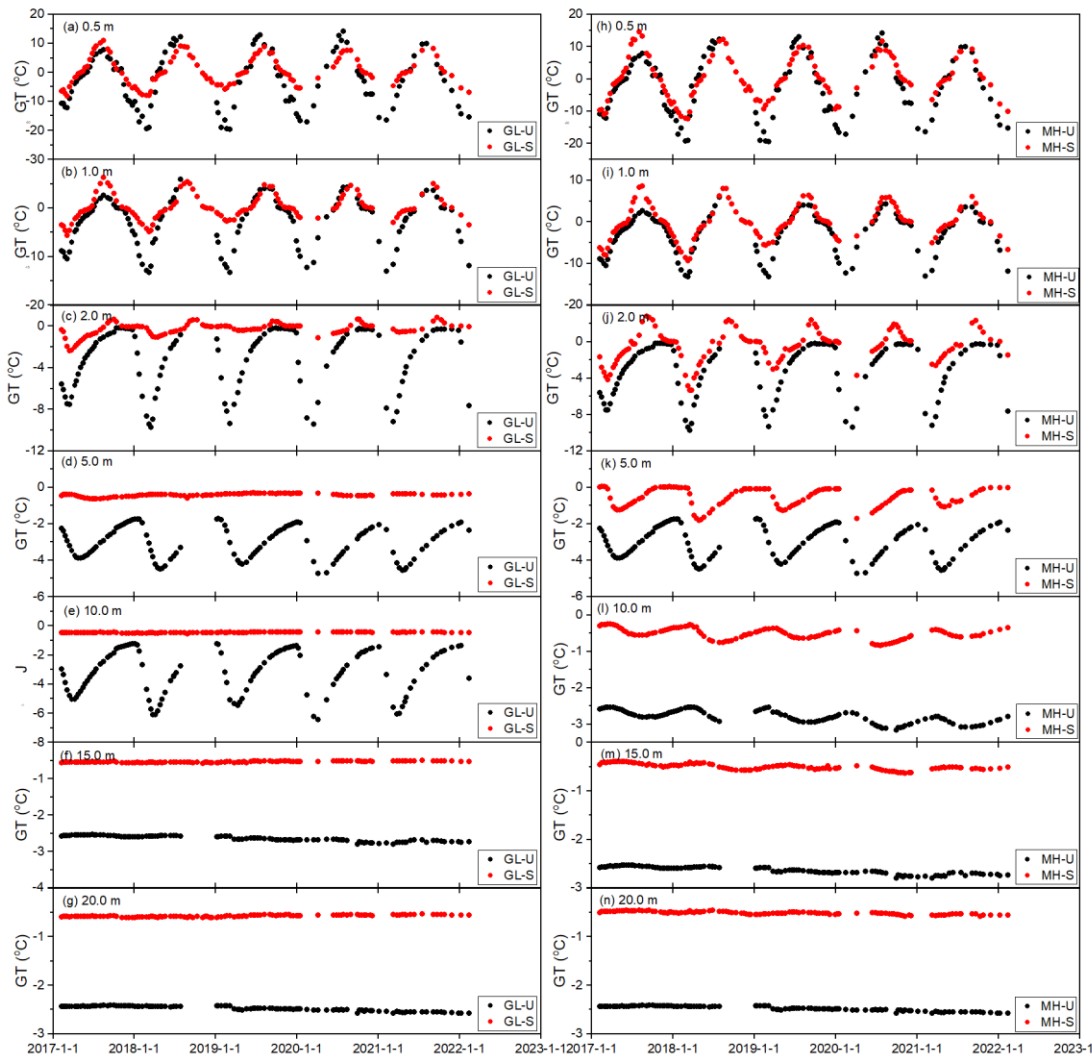


Figure 5. Variations in ground temperatures at depths of 0-20 m at shrub wetlands sites in Gulian

(GL) and Mo'he (MH) on the western flank of the northern Da Xing'anling Mountains in Northeast

China during the period from 2017 to 2022.

Notes: The symbol U stands for the unburned site; S, for the severely burned site, and; GT, for

ground temperature. Figures 5a to 5g were changes in ground temperatures in GL 15-20 years after

fire; Figures 5h to 5n, those in MH 30-35 years after fire.

**3.2 Changes in MAGTs at the permafrost table (MAGT$_{PT}$) and D$_{ZAA}$ (MAGT$_{DZAA}$)**

MAGTs at the permafrost table (MAGT$_{PT}$) and at the D$_{ZAA}$ (MAGT$_{DZAA}$) can truly

reflect the changing characteristics of permafrost thermal regimes. Therefore, in this

section, we have chosen MAGT$_{PT}$ and MAGT$_{DZAA}$ to briefly analyze changes in ground

thermal regimes. When the temperature probe was missing at the actual depth of the

permafrost table or the $D_{ZAA}$, $MAGT_{PT}$ and $MAGT_{DZAA}$ were derived from
interpolation of adjacent ground temperatures.

At the eight monitored sites, the burial depths of permafrost table ranged between

1.5 and 4.5 m, and the $D_{ZAA}$ between 10 and 16 m. From 2017 to 2022, except for GL-
U, MH-U and MH-S sites, $MAGT_{PT}$ and $MAGT_{DZAA}$ decreased gradually ($-0.02$ to
$-0.06°C$/yr), while at other sites increased at rates of 0.01-0.54°C/yr (Figure 6). The
ground warming rates of $MAGT_{PT}$ and $MAGT_{DAZZ}$ were highest at the MG-S site (both
at 0.54°C/yr), and lowest at the GL-S site (0.10 and 0.01°C/yr) (Figures 6a and 6b).
From 2017-2022, the highest differences in $MAGT_{PT}$ and $MAGT_{DAZZ}$ were 2.6 and 1.3°C
at the MG-S site, respectively, and the lowest were 0.2 and 0.1°C at MH-S and AL-S
sites, respectively (Figures 6a, 6d and 6h).

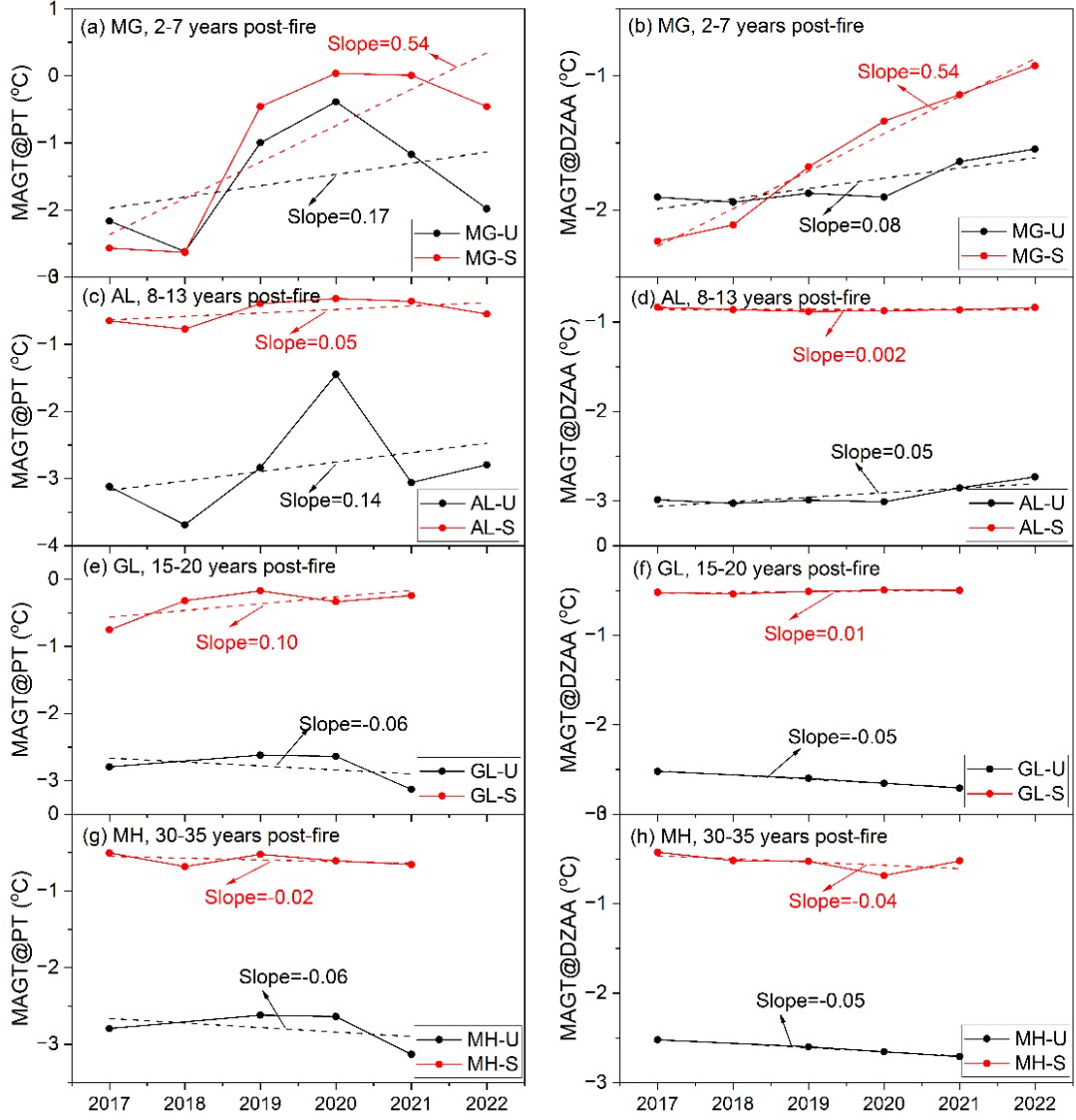


Figure 6. Variations in mean annual ground temperatures at the permafrost table ($MAGT_{PT}$) and the
depth of zero annual amplitude ($D_{ZAA}$) ($MAGT_{DZAA}$) at eight sites in the four study areas (Mangui
or MG, Alongshan or AL, Gulian or GL, and Mo'he or MH) on the western flank of the northern
Da Xing'anling Mountains in Northeast China during 2017-2022.
Notes: U stands for unburned sites, and; S, severely burned sites.

### 3.3 Active layer thickness (ALT) data

ALT, defined as the annual maximum depth of seasonal thaw penetration, was determined according to the deepest position of the 0°C isotherms in a year. Although some data were missing, the change trends of ALT were still obvious (Figure 7).

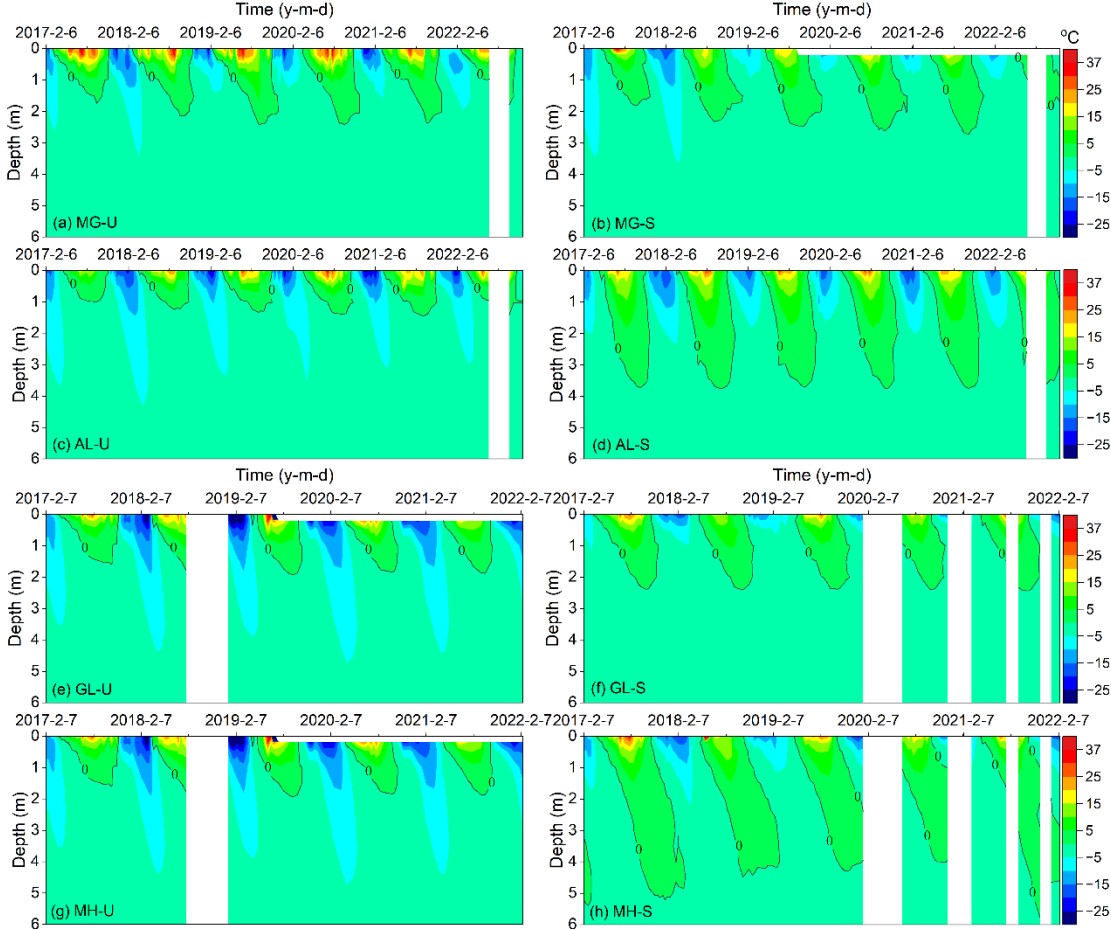

Figure 7. Variability of ground temperatures isotherms at eight sites in Mangui (MG), Alongshan

(AL), Gulian (GL), and Mo'he (MH) on the western flank of the northern Da Xing'anling Mountains

in Northeast China during 2017-2022.

Notes: U stands for the unburned sites, as in insets a (site MG-U), c (site AL-U), e (site GL-U), and

g (site MH-U), and S, the severely burned sites, as in insets b (site MG-S), d (site AL-S), f (site GL-

S), and h (site MH-S).

ALT was between 1.0 and 5.2 m at the eight sites from 2017 to 2022, and the

maximum average of ALT was 4.5 m at MH-U and the minimum was 1.6 m at AL-U.

Compared with the other seven sites, MH-S has the largest ALT, with the maximum

ALT at 5.2 m in 2017. From 2017 and 2022, only at the MH-S site, ALT decreased at a

rate of 36.5 cm/yr, while at the other sites it increased at rates of 0.1-20.5 cm/yr. The

increase rate of ALT at MG-S was the fastest, and; at AL-S, the slowest (Figure 8).

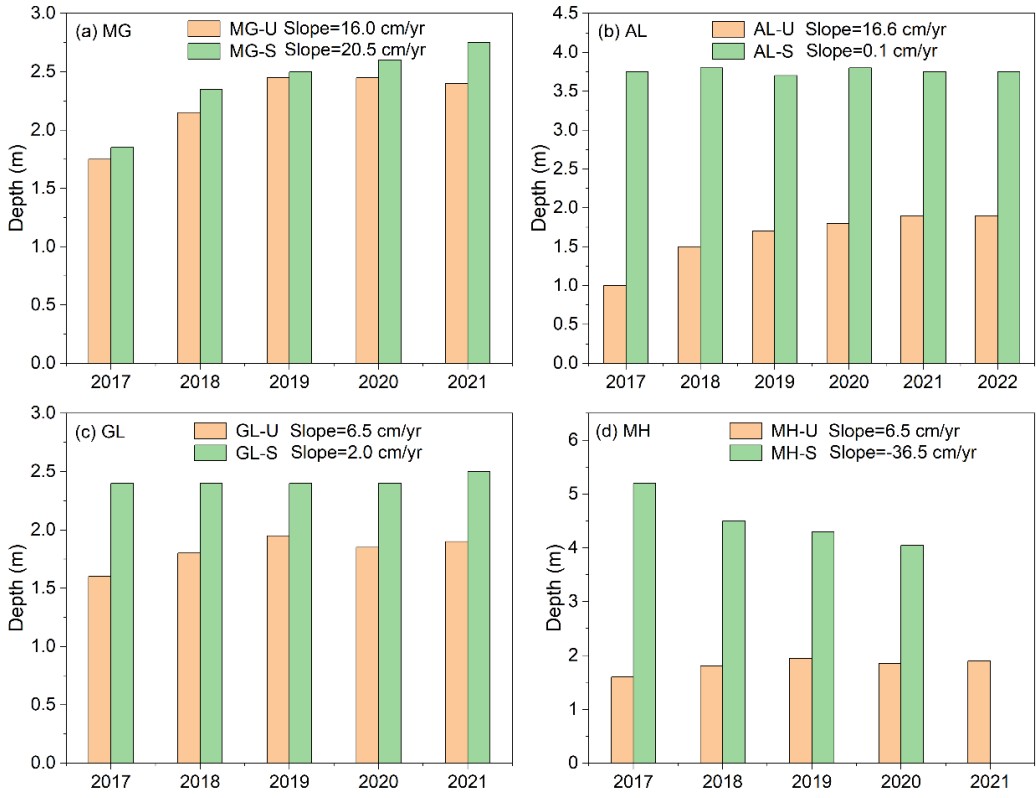

Figure 8. Variation characteristics of active layer thickness (ALT) from 2017 to 2022 at eight sites of the four study areas in Mangui (MG), Alongshan (AL), Gulian (GL), and Mo'he (MH) on the western flank of the northern Da Xing'anling Mountains in Northeast China during 2017-2022. Notes: U stands for the unburned site, and S, the severely burned site.

**3.4 Variations in gravimetric soil moisture content (SMC)**

At MG-U and AL-U sites, SMC decreased with increasing depth, especially in the active layer and near-surface permafrost, or in the vicinity of the permafrost table (Figure 9). For example, at AL-U, SMC decreased at a rate of 8.6%/m and average SMC was 108.2±11.7% at depths of 0-9.4 m (Figure 9b). At the depths (0-3 m) with higher SMC, the soil contains massive ice crystals and a large amount of segregated ice, with ice lenses of 0.1–5.0 cm in thickness. For example, at GL-U, SMC was higher at the junction of the bottom of the active layer and the upper layer of transient permafrost (1-2 m in depth) due to a large amount of segregated ice (0.2-5.0 cm thick) immediately under the permafrost table. At MG-S, AL-S, GL-S, and MH-S sites, changes in SMC

were inconspicuous, only at depths of 0-0.5 m, with a slight decreasing trend. At depths
of 0.5-9.4 m, differences in SMC were minor (Figure 9). At MG-S, SMC fluctuated
between 11.7-63.2% at depths of 0.6-7.6 m, with average SMC at 27.5±3.2% (Figure
9a). At AL-S, GL-S, and MH-S sites, SMC fluctuated between 4.7-26.6% at depths of
0.6-8.4 m, with average SMC of 17.1-21.1%.
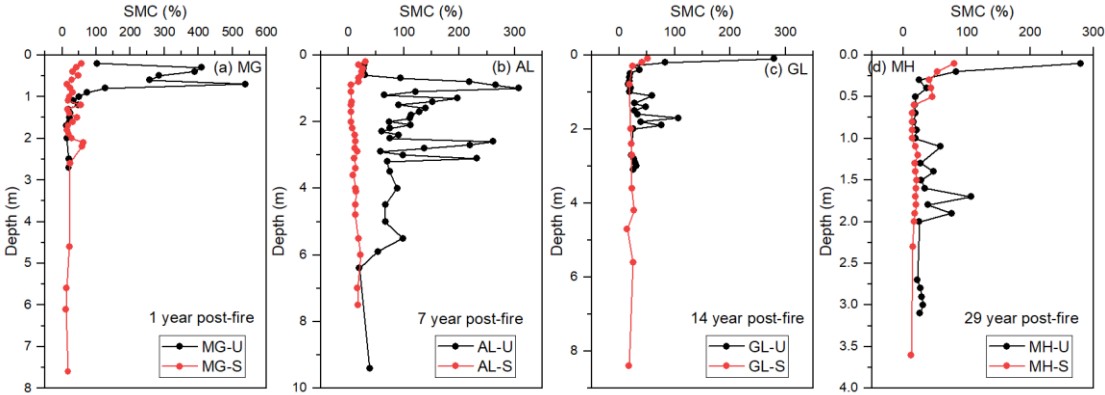
Figure 9. Variations in gravimetrically-based soil moisture contents (SMC) with different fire
severity at eight sites in Mangui (MG), Alongshan (AL), Gulian (GL), and Mo'he (MH) on the
western flank of the northern Da Xing'anling Mountains in Northeast China in 2016. Notes: The
symbol U stands for unburned, S for severely burned, and; SMC, for soil gravimetric moisture
content.
**3.5 Variations in soil nutrients**
The contents of SOC and TN decreased with increasing depths. A large amount of
SOC and TN were stored in the active layer (0-1.3 m), especially in the soil organic
layer (0-0.5 m) (Figures 10a to 10n). The change trends of SOC and TN were consistent.
For example, at MG-U, at depths of 0-1.3 m, averages of SOC and TN were 140.5±26.9
and 5.9±0.9 g/kg, respectively; at depths of 1.3-2.5 m, changes in SOC and TN were
relatively smooth, fluctuating between 2.0-13.3 and 0.9-1.5 g/kg, with averages at
5.4±1.1 and 1.2±0.1 g/kg, respectively (Figures 10a and 10b).

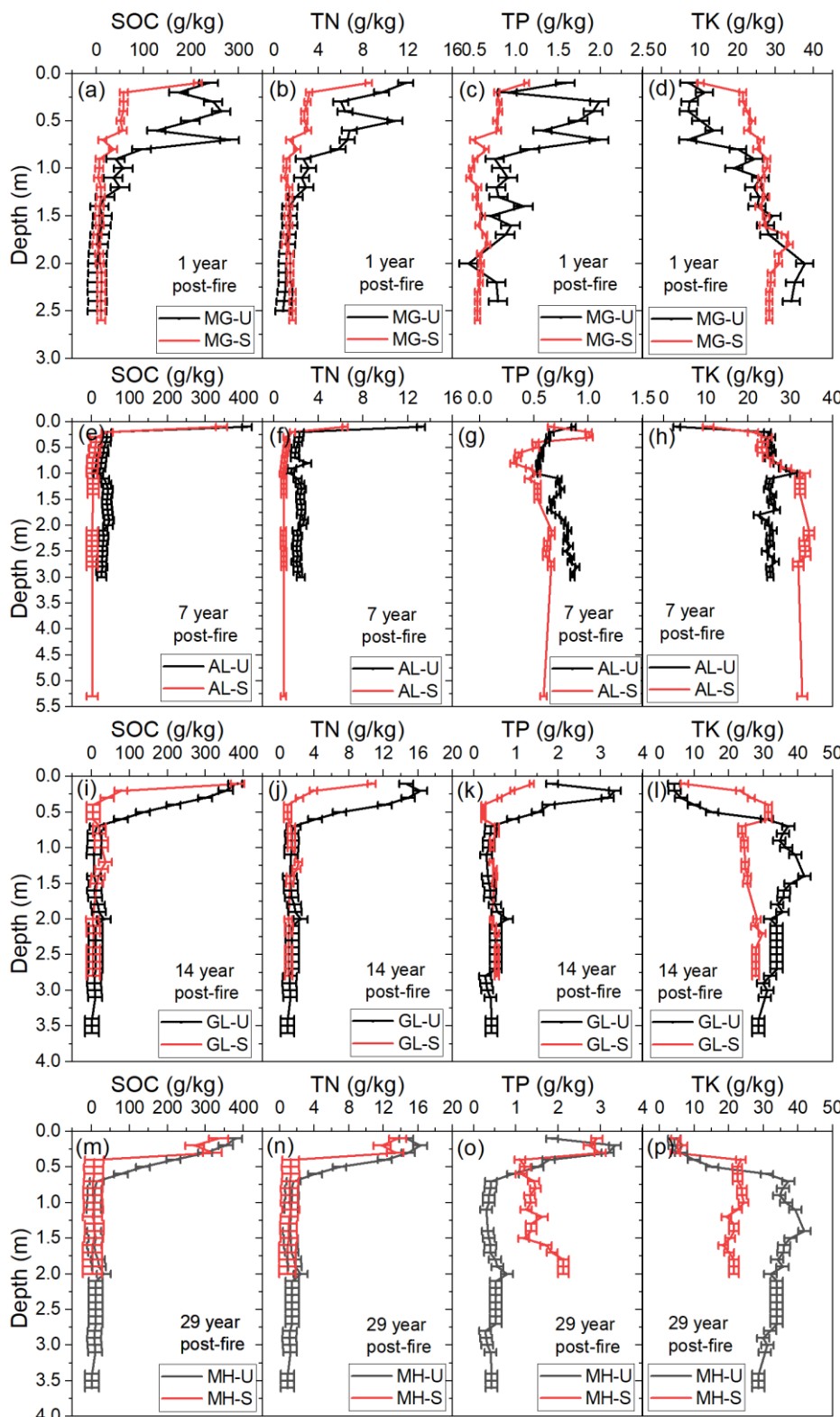


Figure 10. Variations in soil nutrients at eight sites in Mangui (MG, a to d), Alongshan (AL, e to
h), Gulian (GL, i to l), and Mo'he (MH, m to p) on the western flank of the northern Da Xing'anling
Mountains in Northeast China in 2016.

Notes: The symbol U stands for unburned, and S for severely burned. SOC stands for soil organic
carbon; TN, for total nitrogen; TP, for total phosphorus, and; TK, for total potassium.

TP contents decreased up to 1.0 m in depth, and changes in TP were minor at depths of 1.0-5.3 m (Figures 10c, 10g, 10k, and 10o). For example, at MG-S, TP decreased at a rate of 0.56 g/kg/m at depths of 0-1.0 m, with an average of 0.7±0.1 g/kg (Figure 10c); TP fluctuated between 0.4 and 0.7 g/kg at depths of 1.1-2.6 m, with an average of 0.6±0.01 g/kg. The change trends of TK were opposite with TP because TK contents increased downwards (Figures 10d, 10h, 10l, and 10p). The contents of TK were all below 41.8 g/kg. For example, at MG-U, TK increased at a rate of 14.1 g/kg/m, while TP decreased at a rate of 0.5 g/kg/m (Figures 10c and 10d).

## 4. Data availability

The dataset of ground temperature, ALT, SMC, SOC, and contents of TN, TP, and TK can be freely downloaded and is available from the National Tibetan Plateau/Third Pole Environment Data Center (https://doi.org/10.11888/Cryos.tpdc.300933, Li and Jin, 2024). The dataset was classified into three categories: ground temperatures (at MG-U, MG-S, AL-U, AL-S, GL-U, GL-S, MH-U, and MH-S), soil moisture contents (SMCs), and soil nutrient contents (SOC, TN, TP, and TK).

## 5. Conclusions

The Da Xing'anling (Hinggan) Mountains in Northeast China are located on the southern margin of the Eastern Asia permafrost zone and boreal forest belt. It is an area where fires occur frequently and the thermal state of permafrost is sensitive to fire disturbances. To study fire effects on the permafrost environment, a monitoring network has been established in Northeast China since 2016. Therefore, a long-term field dataset on ground hydrothermal regimes and soil nutrients has been obtained. This dataset fills a gap in a monitoring study of fire effects on the permafrost environment in the hemiboreal forest zone in Northeast China. These data include ground temperatures at depths of 0-20 m, SMC at depths of 0-9.4 m, and contents of SOC, TN, TP, and TK at depths of 0-3.6 m. The data were collected from eight sites in four burned areas (MG in Mangui, AL in Alongshan, GL in Gulian, and MH in Mo'he) with two categories of fire severity (severely burned and unburned) from 2016 to 2022.

Long-term monitoring data in the northern Da Xing'anling Mountains in Northeast

China have shown a degrading permafrost under the disturbances of climate change and frequent forest fires. This is evidenced by rising ground temperature, thickening active layer, and evidently changing SMC and soil nutrient contents. The 6-year long dataset presented in this study has a high-quality time series with only a few missing data. This valuable and hard-won dataset of forest fires and permafrost is worth maintaining and improving in the future. This study provides important basic data for the protection of the ecosystem-dominated Xing'an permafrost and herewith boreal permafrost ecosystems. Furthermore, it is useful for more accurate prediction of fire-induced permafrost changes and for more accurate estimating and better-managing soil carbon stocks. It also provides an important reference for the initiatives of carbon neutralization and carbon peaking control and the assessment of infrastructure safety under fire threats.

**Author contributions.** XL and HJ designed and conducted this research. XL compiled the dataset, performed the data analysis, and wrote the manuscript. RH, HW, XC, RŞ, and ZT participated in the fieldwork. HJ, QF, QW, DL and RŞ improved the writing. XL prepared the manuscript with contributions from all co-authors.

**Competing interests.** The authors declare no conflict of interest.

**Disclaimer.** Publisher's note: Copernicus Publications remains neutral with regard to jurisdictional claims in published maps and institutional affiliations.

**Acknowledgments.** We would like to thank all the scientists and students who participated in the fieldwork. We thank the two anonymous reviewers and editors for their thorough reviews and insightful comments that improved the paper. We also are grateful to Professor Xin Li for his encouragement, guidelines, and review of the proposal for writing up this paper and preparation of the datasets.

**Financial support.** This research has been supported by the National Natural Science Foundation of China (Grant Nos. 42471166 and 32241032); the program of the Key Laboratory of Cryospheric Science and Frozen Soil Engineering, CAS (Grant No. CSFSE-ZQ-2407); Heilongjiang Excellent Youth Fund (Grant No. YQ2022D002),

National Natural Science Foundation of China (Grant No. 42101408), and; Fundamental Research Fund for the Central Universities (Grant Nos. 2572023CT01 and 2572021GT08). Raul-David Șerban received funding from the Autonomous Province of Bozen/Bolzano-Department for Innovation, Research and University (Grant No. 13585/2023).

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
