# Peer review of "An integrated dataset of ground hydrothermal regimes and soil 1 nutrients monitored during 2016-2022 in some previously burned 2 areas in hemiboreal forests in Northeast China 3 Xiaoying Li1, Huijun Jin1,2,3\*, Qi Feng4, Qingbai Wu1, Hongwei Wang1,"

_Earth System Science Data, 2024_

## Author Response (AR1)

This manuscript describes a six-year time series of permafrost temperature observations at four sets of paired burned/unburned sites in northeastern China. In addition, the data set includes analyses of gravimetric moisture content as well as soil nutrients derived from incremental soil/permafrost cores collected when the boreholes were installed. This is a relatively straight forward data set - similar to others I have seen published directly at data repositories without an accompanying data publication. Overall, this is a useful and informative data set for understanding the influences of fire on permafrost temperature. Before the manuscript can be considered further for publication a number of clarifying improvements are necessary.

**Response:** Thank you for all your efforts for and time spent on our manuscript.

These suggestions and reviews are very helpful for improving the quality of our manuscript. On the basis of your suggestions, I made some modifications to this manuscript as advised.

While the introduction is useful, I find it a little broad, and think it would be helpful to have more focused contextual information. For example, is this an under sampled location compared to areas within the northern permafrost zone? Are there similar sites nearby in the GTN-P? It would be nice to know a little about the permafrost conditions of the region earlier in the manuscript. Is this ice rich permafrost? Similarly, basic information on the fire regime and any recent changes would be helpful (from the literature). One of the sites shows active layer recovery after fire, without context it is difficult to tell if this is an anomaly, or something to be expected more broadly. Some of this information is provided, but it is scattered throughout the manuscript.

**Response:** Agreed and done.

According to your suggestions, we have revised the ***Introduction***. This is an area that is under-sampled and has little data sharing compared to the Arctic permafrost region. In the GTN-P monitoring network, the monitoring of permafrost changes after forest fires is also very rare.

Permafrost is generally warm and thin in Northeast China. In this study, the unburned site is underlain by tice-rich permafrost at 0-3 m in depth. Relevant information on fire regimes and any recent changes is given in the ***Introduction***. The characteristics of variations in ground hydrothermal state and active layer thickness (ALT) after forest fire are also described.

Please check the texts in Lines 56-61, 71-83, 93-102, and 108-116 for more details.

The following comments point to specific issues as well as more minor editorial areas for improvement.

L24: typo - succession

**Response:** Agreed and done.

Changed to the "succession" in Line 24.

L58: It would probably be good to specify the "soil organic layer"

**Response:** Agreed and done.

Changed the "organic layer" to the "soil organic layer". In Line 62.

L65-75: This is all accurate, however it is probably worth noting that the depth of the seasonally thawed active layer has been observed to decline with ecosystem recovery after fire (e.g. Rocha et al 2012).

Rocha, A. V., Loranty, M. M., Higuera, P. E., Mack, M. C., Hu, F. S., Jones, B. M., et al. (2012). The footprint of Alaskan tundra fires during the past half-century: implications for surface properties and radiative forcing. Environmental Research Letters, 7(4), 044039. https://doi.org/10.1088/1748-9326/7/4/044039

**Response:** Agreed and done.

In Lines 80-83, we have added the sentence

"*Moreover, changes in ground hydrothermal regimes and ALT would decline and progressively dwindle with ecosystem recovery and organic layer regrowth over time under a stable or cooling climate (e.g. Holloway et al., 2020; Rocha et al., 2012).*"

L86-87: Is this assertion for ecosystem protected permafrost, or all cases of permafrost under the impacts of wildfire? I would suspect the former. Ice content is probably also important as well.

**Response:** Yes, it was originally emphasized the "ecosystem-protected permafrost". But according to another reviewer's suggestion, it should refer to all permafrost types. So the "ecosystem-protected permafrost" was deleted, and now it refers to all cases of permafrost: ecosystem driven, protected or modified, or transitory types.

In addition, we agree with you that ground ice content is very important. Because in the case of high or very high ice contents, permafrost degradation or thaw will result in significant thermokarst landforms, such as thaw slumps, thermokarst lakes, and others. The formation of these thermokarst landforms is conducive to carbon release.

L94: How can permafrost be prone to wildfire?

**Response:** This expression needs some modifications. It should be

"*Therefore, this ecosystem-dominated (driven, modified, or protected) permafrost is sensitive to climate warming and wildfires (Shur and Jorgenson, 2007).*"

Deleted the "prone to" (Lines 110-111).

L110: Are soil nutrient contents being continually observed?

**Response:** Not, they are not.

Soil nutrients were only measured in 2016, rather than being continuously observed. Thus, changed the "a continuous observation system" to the "an observation system". In Line 131.

**L115: Should this be section 2?**

**Response:** Agreed and done. Changed to the "In **Section 2**". In Line 137.

L138: It would be good to describe the fire severity sampling before this point. I found myself wondering if and how severity was incorporated in site selection.

**Response:** Agreed and done.

Placing the "**Section 2.2 *Fire severity***" section before the "**2.1 *Study area descriptions and monitoring networks***" seems wrong and reverse in the order, so to make this sentence more clearly, we have changed this sentence to:

" *The network includes eight sites in the four burned areas with two fire severity (severely burned (S) and unburned (U)) from 1987 to 2015 (the fire severity division method was shown in "**2.2 Fire severity**" section))*." (Lines 158-161).

L141: What are the units hm^2?

**Response:** The $hm^2$ is the hectare (ha), and changed to the "ha".

L201-203: Are these values appropriate for this forest type?

**Response:** Through the method of this literature (Cocke et al., 2005), this value is suitable for this type of forest, and the fire severity of MG area divided by field investigation is consistent with the result of this method.

L204-205: This is a curious statement, why would damage/device malfunction be more prevalent in moderate burned relative to severe and unburned?

**Response:** Since most of the moderate burned sites were located far from the roadside, it was difficult for the rigs to enter into the forest, and; the instruments at the light burned sites were basically destroyed by curious people  because these sites are relatively close to the road. Thus, most of the data at these sites were missing. Therefore, in order to ensure the uniform fire severity and data integrity, only severe burned and unburned sites were selected for all areas.

L211: Isn't it just burned and unburned?

**Response:** Agreed and done.

Changed to the "At each site of unburned and severe burned" in Line 236.

L222: In looking at the data it seems like three times per month is maximum - and one or two times per month is more common. Line 262 indicates collection occurred 1-2 times/month after the COVID-19 pandemic

**Response:** Yes, the original design of data collection was thrice monthly, and most data collection was guaranteed thrice monthly before the COVID-19 pandemic. However, due to the impacts of weather and road damage, more collection was only made to be twice monthly, and; Once monthly was also rare. However, during the outbreak of the COVID-19 pandemic, due to the resultant local restrictions of personnel movement/traffic, we tried our best to ensure the collection of data thrice monthly. Thus, most of the time, the data collection was made twice or once monthly.

This sentence is not properly expressed. Thus, we have changed this sentence to

"*Since February 2017, ground temperatures at these boreholes were manually measured thrice monthly (Table 2), or occasionally once or twice monthly due to traffic difficulty or control, by a multi-meter Fluke 189® device.*" (Lines 246-248)

L235: Table 2 is a little unclear - it seems that soil moisture and nutrients are one-time observations, and the only thing being monitored over time is temperature. This distinction should be made clearly - I'm not sure that this table is warranted.

**Response:** Agreed and done.

Yes, it is one-time observations for soil moistures and nutrients in 2016, and thrice monthly for ground temperatures in 2017-2022. Thus, we have changed the table and added the table notes immediately under Table 2.

L253: How was SE calculated? Were there multiple sub-samples analyzed from each depth?

**Response:** At the time of sampling, we made three replicas in the same soil layer and mixed the three samples into one, so there was only one datum at each depth, and then calculated the SE from the data at all depths.

SE=Standard deviation/$n^{1/2}$, n is the number of samples.

Adde the sentence

"*Three replicas were collected at the same depth and then three samples were evenly mixed into one.*" in Lines 269-270.

L254: How is time quantified? This data set seems somewhat small given that there are two different ecosystem type.

**Response:** Because it is very difficult to achieve decades of permafrost observation in the same burned area. Therefore, we selected these areas with very similar vegetation and permafrost types and landforms to observe changes in the permafrost environment in several years to decades after fire using the (chronosequence) approach, i.e., "spatial variations as a substitute for temporal changes". For example, in the larch forest, the change of permafrost in MG area was 1-7 years after fire, and in AL was 7-13 years after fire, so that we can know the change trend of permafrost 1-13 years after fire.

L259: Should be pandemic, since it was not restricted to a specific geographical region.

**Response:** Agreed and done. Changed to the "pandemic".

L341: In figure 7 it might be easier to compare burned and unburned for each site if they were side by side (i.e. a single row), rather than above/below each other.

**Response:** Agreed and done.

Figure 7 was redrawn.

[Figure]

Figure 7. Variability of ground temperatures isotherms at eight sites in Mangui (MG), Alongshan (AL), Gulian (GL), and Mo'he (MH) on the western flank of the northern Da Xing'anling Mountains in Northeast China during 2017-2022.
Notes: U stands for the unburned sites, as in insets a (site MG-U), c (site AL-U), e (site GL-U), and g (site MH-U), and S, the severely burned sites, as in insets b (site MG-S), d (site AL-S), f (site GL-S), and h (site MH-S).

**Anonymous Referee #2**

This paper presents 6 years of data in four areas of Northeast China at paired burned and unburned sites, including ground temperature, soil moisture, and soil nutrients. The four areas represent a chronosequence of time since fire, allowing for interpretation of the impacts of fire on permafrost over time.

The dataset is not new or unique (there is similar data from Canada and Alaska), but the location is new as there is little previous work on the impacts of fire on permafrost in China. The methods are not new, most are standard for permafrost science. I am surprised that the authors did not use a temperature logger that stored hourly measurements (e.g. Onset HOBO U23 Pro v2 Temperature/RH Data Loggers) as that would've eliminated the need for weekly visits to the sites and would've allowed them to collect much more data with fewer gaps.

**Response:** Thank you very much for your suggestion about using a temperature logger that stored hourly measurements. In the initial data collection, the use of temperature logger was not considered. However, due to the COVID-19 pandemic in 2020-2023, the data could not be collected on time, and the cost of collection was also considered. Therefore, in 2023, we installed temperature loggers to automatically collect data every hour. However, unfortunately the temperature loggers were damaged by some curious people, resulting in lost temperature loggers and battery. In addition, since there was no network signal in the study area, the data could not be transmitted wirelessly. Therefore, when it was found that some time had passed since, which eventually led to the loss of some important data. At present, we are also thinking of a better solution to avoid the temperature logger's destruction and data loss as much as possible.

The dataset is complete, and I was able to access and download it from the given identifier. It is useable in its current format and size. I do feel that the data could be useful in the future, particularly for those conducting modelling studies or as a baseline for future changes in permafrost conditions. There is one inconsistency within the data, see my major comment below, so I'd like the authors to provide an explanation for that. I do not feel that the meta data is sufficient unless accompanied by the current manuscript under review. I would like to see more information about the sites and the data collected in the metadata, as well as an explanation for the data gaps.

**Response:** Meta data are automatically generated by the data website and cannot be changed by ourselves. Thus, I have added a metadata description to upload to the data list.

The article length was appropriate and it was well structured. There were some grammatical errors throughout, but especially in the conclusion. The figures and tables were good but overall not enough detail in the captions (see minor comments below for specific places).

**Response:** Thank you very much for your suggestions. We have corrected the grammatical errors of the whole article and given a more detailed description of the figures and tables captions.

 Major comments:

The introduction is quite general. For example, I don't feel that it is relevant to mention tundra fires as your data is for the hemiboreal environment. Your focus is on hydrothermal regimes and nutrients, but you didn't give any background on post-fire permafrost soil moisture literature or any nutrients other than carbon.

**Response:** According to your suggestions, we have revised the ***Introduction***. The contents about tundra fire had been deleted, and the background information about the hydrothermal state and nutrients changes of permafrost after fire had been added. In Lines 71-83, 93-102.

*"In the boreal zone, 6-11 years after fire, mean annual ground temperature (MAGT) increased by 1.5-2.3ºC (Li et al., 2021; Munkhjargal et al., 2020; Nossov et al., 2013; Smith et al., 2015), even mean annual ground surface temperatures in burned areas were still 2-3ºC higher than that in unburned areas 80 years after fire (Brown et al., 2015). Meanwhile, 25 years after fire, the active layer thickness (ALT) could increase by 2.75 m, and ALT could not recover to the pre-fire level even 36 years after fire (Viereck et al., 2008). In Central Siberia, it generally takes 70-80 years for the active layer to return to the pre-fire state (Kirdyanov et al., 2020). In addition, forest fires result in decrease in soil moisture content, which in turn affects ground thermal regimes (Nossov et al., 2013). Moreover, changes in ground hydrothermal regimes and ALT would decline and progressively dwindle with ecosystem recovery and organic layer regrowth over time under a stable or cooling climate (e.g. Holloway et al., 2020; Rocha et al., 2012)."*

*"In addition to soil organic carbon, forest fires potentially also reduce soil nitrogen and phosphorus stocks, inducing shifts in nutrient cycling (Certini, 2005; Gu et al., 2010; Knicker, 2007). For example, one year after wildfire in interior Alaska in the boreal zone, soil carbon content was about 1071-1420 g/m² less at the sites of burned soils than that of unburned soils, and; burned soils had lower nitrogen than unburned soils, higher calcium, and nearly unchanged stocks of potassium, magnesium, and phosphorus (Neff et al., 2005). As a result, wildfires in boreal forest had been considered to trigger strong positive feedbacks on climate warming via massive emissions of biogenic major greenhouse gas (Koven et al., 2015; Ramm et al., 2023)."*

You're using a chronosequence, but I find it hard to know which sites are where in the sequence. Maybe a schematic showing time since fire and then the names of the sites would help. It will help the reader be able to interpret the results better, especially as time since fire is extremely important to infer post-fire impacts on permafrost. For example, Figure 3 is interesting to me, particularly because 3 of the areas show differences in MAGT between the burned and unburned sites at depths of 20m, except for MG. I'm wondering if this is because MG is the end of the chronosequence, decades after fire, and things have returned to pre-fire conditions, but I had to scroll around and find it on page 8. I think in general more attention needs to be paid to this in the paper. All the results should be interpreted with this in mind, which is currently lacking in the paper.

**Response:** Agreed and done.

According to your suggestions, we have marked the post-fire time on the figures. For MG, it is the early stage of the fire, and the effect of the forest fire on the soil temperature had not yet reached the depth of 20 m. Thus, the difference between the burned and unburned sites was relatively small. At the time of the initial submission,

we analyzed the data in chronosequence, but according to the editor's request and suggestion, this paper is a data description article and should refrain from making data interpretations,. Asa result, the analysis and interpretation of the data had been deleted and only described the changes in the data. However, in order to help the reader to better interpret the results, the post-fire time has been added to the figures according to your comments.

Minor comments:

Abstract

Line 31: I wouldn't consider a 6-year dataset to be long-term.

**Response:** Agreed and done. Changed "long-term datasets" to the "The datasets". Line 31

Introduction

Line 78: There are many other references about wildfire, permafrost and carbon. Some examples:

O'Donnell JA, Harden JW, McGuire AD, Kanevskiys MZ, Jorgenson MT, Xu X. The effect of fire and permafrost interactions on soil carbon accumulation in an upland black spruce ecosystem of interior Alaska: implications for post-thaw carbon loss. Glob Chang Biol. 2011;17:1461-1474.

Genet H, McGuire AD, Barrett K, Breen A, et al. Modeling the effects of fire severity and climate on active layer thickness and soil carbon storage of black spruce forests across the landscape in interior Alaska. Environ Res Lett. 2013;8(4):045016.

O'Donnell JA, Harden JW, McGuire AD, Romanovsky VE. Exploring the sensitivity of soil carbon dynamics to climate change, fire disturbance and permafrost thaw in a black spruce ecosystem. Biogeosciences. 2011a;8:1367-1382.

Dieleman C.M., Day N.J., Holloway J.E., Baltzer J., Douglas T.A., Turetsky M.R. 2022. Carbon and nitrogen cycling dynamics following permafrost thaw in the Northwest Territories, Canada. Science of the Total Environment, 845(1), 157288. https://doi.org/10.1016/j.scitotenv.2022.157288

**Response:** Agreed and done.

We added these citations at the end of the sentence as advised. Check citations in Lines 85-86 and added references in *References*.

Line 82: What about permafrost that is not ecosystem-protected? What about low ice-content sites that don't experience thermokarst or "abrupt thaw"? Much of the boreal forest that is of this type and would still release carbon post-fire. I think it's important to not focus only on ecosystem-protected or sites prone to thermokarst (unless your data applies only to those settings, then I think much more detail would be required in this introduction).

**Response:** Agreed and done.

Permafrost that is not ecosystem-protected and areas with low ice-content and that does not experience thermokarst or "abrupt thaw" also release large amounts of carbon after permafrost degradation. These are suitable for different types of boreal permafrost. The original purpose of this sentence was to emphasize the permafrost in this study area. The permafrost in Northeast China belongs to the ecosystem-dominated permafrost, but it is also the unstable permafrost, which will degrade rapidly or even thaw completely under the joint disturbances of climate change and forest fire. Thus, we have made some modifications as advised. Lines 94-102.

Line 104: Often complex how?

**Response:** Unlike tundra, which has less variations in topography, the terrain in the northeastern Da Xing'anling Mountains, NE China is more complex. Thus, forest fires can occur in gullies, hillsides and hilltops. Meanwhile, forest fires mainly occur in spring and autumn in Northeast China, but there are more fires in spring. Due to the low precipitation and dry climate in spring, the ground surface litter layer is thick and dry. There is more precipitation in autumn, the snow accumulation period begins, and the leaves are moist. Summer precipitation is concentrated, if there are forest fires, most of them are lightning fires. Winter is covered by thick snow, so fires are rare. Therefore, the occurrence of fires in the Da Xing'anling Mountains has strong seasonality.

Section 2

Line 138: How was fire severity classified? What does "severely burned" indicate (i.e., a proportion of the organic layer lost, entire destruction of the canopy, etc.)?

**Response:** Fire severity is divided according to Lines 216-233.

Fire severity can be divided according to the loss of organic layer and forest canopy immediately after fire. Several years after fire, the fire severity should be classified according to the differential Normalized Burn Ratio (dNBR) calculated from the remote sensing images of the fire areas in the same year of fire.

Figure 1: It's unclear to me where the permafrost region is in (a). More description is needed in the figure caption. Are the pink areas in the Landsat images burned areas?

**Response:** The light blue areas in Figure 1a is the permafrost region. Figures 1c to 1f were the false-color composite image of the remote sensing image, the pink areas are the burned area, and the green area are the unburned areas. These descriptions are also added to the figure captions. Lines 170-173.

Line 161: Why were these locations selected in particular?

**Response:** Because forest fires in the Da Xing'anling Mountains, NE China often occur in the primeval forest and cannot be easily accessed by walking or driving. Therefore, in order to observe the permafrost environment in the burned areas, we need to select the areas that can be reached on foot or using vehicle, so that the drilling rig could enter the research sites and drill the observation holes of soil temperatures. Secondly, in order

to observe long-term post-fire changes in the permafrost environment, we choose the burned area from 1987 to 2015, so that the chronosequence can be used to observe longer-term changes of permafrost after fire. In addition, the selected areas have consistent topography and vegetation types.

Line 172: How do you know the pre-fire organic layer thickness? Organic layers vary substantially over short distances, I'm surprised the window for the site is only 5cm pre-fire, unless it was only measured in one spot.

**Response:** The organic layer thickness data here are all measured values in this study areas (four areas). In the vicinity of the burned areas of several meters to tens of meters, we selected the unburned area as the control site, that is, the unburned site in the paper. The organic layer thickness of the unburned sites and the burned sites were measured in 2016. The organic layer in the unburned areas in the Da Xing'anling Mountains, NE China is relatively thick, generally between 40-60 cm. That in the burned area is much thinner.

Table 1: A relatively large amount of organic layer remains after the fires at all of the site (minimum 20 cm). I think it's important to note this somewhere in the paper, as this minimizes post-fire changes (less active layer thickening and ground temperature increase) than if, for example, less than 5 cm remains.

**Response:** We have mentioned relevant information about the thickness of organic layer in the appropriate places in the article. We have added this sentence

"*Moreover, changes in ground hydrothermal regimes and ALT would decline and progressively dwindle with ecosystem recovery and organic layer regrowth over time under a stable or cooling climate (e.g. Holloway et al., 2020; Rocha et al., 2012).*" in Lines 80-83.

Line 189: Why do you provide less details for the GL sites than the other sites? How far apart were they, when did measurements commence and how long after fire?

**Response:** Lines 186-187 are also descriptions of GL site, not only the Line 189, including distances apart they and the measurement time. The description of these sites is the same.

Line 201-203: Why were these thresholds chosen?

**Response:** This is the common method of international fire severity division, and it is also a standard means of division. According to the Cocke et al., (2005) and Roy et al., (2006), the dNBR optimality values for these average changes are 0.241 for grass and 0.57 for shrub. Therefore, these values are selected as threshold values through the classification of fire severity by vegetation burn status and the comparison with dNBR (Key and Benson, 2006; Escuin et al., 2008).

Cocke, A. E., Fulé, P. Z. and Crouse, J. E.: Comparison of burn severity assessments using Differenced Normalized Burn Ratio and ground data, Int. J. Wildl. Fire, 14, 189-198, 2005.

Escuin, S., Navarro, R. and Fernandez, P.: Fire severity assessment by using NBR (normalized Burn ratio) and NDVI (normalized difference vegetation index) derived from LANDSAT TM/ETM images. Int. J. Remote Sens., 29(4), 1053-1073, 2008.
Key, C. H. and Benson, N. C.: "Landscape assessment (LA)." FIREMON: Fire effects monitoring and inventory system 164: LA-1, 2006.
Roy, D. P., Boschetti, L. and Trigg, S. N.: Remote sensing of fire severity: assessing the performance of the normalized burn ratio. IEEE Geosci. Remote Sens. Lett., 3(1), 112-116, 2006

Table 2: I'm finding this table hard to interpret. For the last three columns, upon first glance, it looks like they only apply to site AL-S and GL-U. I was wondering why no ground temperature measurements were taken for all the other sites. But I think it is just the way the table is organized. Can you rethink this to make it more clear?

**Response:** Ground temperature measurements were taken for all the other sites. Because the observed depth of ground temperature, time period, and monitoring frequency are the same at all sites, so the tables for all sites are combined into one. For clarity, we have added a specific description to each site based on your suggestions.

Line 238: What are complex changes? Please describe.

**Response:** At depths of 0-3 m, changes in soil moisture content (ice content), organic matter and lithology are the most significantly. Therefore, compared with those at depths of 3-20 m, changes in soil nutrients and ground temperature are more dramatic.

Thus, it was changed the sentence to the

"*At depths of 0-3.0 m, samples were collected every 10 cm in depth in soil strata with more significant changes of soil organic matter and lithology near the ground surface.*" Lines 270-272.

Line 271: How were they quality controlled?

**Response:** During each measurement, the multi-meter Fluke 189® device is used to check whether there are abnormal values in each cable of the ground temperature line, and the cables with abnormal values are measured multiple times and recorded in detail. Record data when the measurement is stable.

Figure 3: Why do all the areas except MG show a large difference between burned and unburned sites?

**Response:** Since MG is only 2-7 years after a fire, soil temperature is in the increasing stage. Form Figure 3a, there is a significant difference between the severe burned site and the unburned site at depths of 0-8 m. With the increase of post-fire time, soil temperature will continue to rise. Thus, the difference in soil temperature of other three areas is larger than that of MG.

Line 296: What dates were annual averages (e.g. MAGT) taken over? Calendar year?

**Response:** The date has been given in the figure caption and is the average from 2017-

2022 (6 years).

Table 3: Similar to Figure 3, it's curious that the unburned MAGT at 0.5 m depth for MG was warmer than the burned. Any explanation for this? This is also shown on Figure 4a and Figure 7a.

**Response:** Because the journal requires as little interpretation of the data as possible, according to the editor's request, the reasons for the data interpretation were deleted.

This is mainly due to the formation of a large number of thermokarst ponds (alas) on the ground surface in summer at severe site in MG, with water depths of up to 10–20 cm in summer, which results in a decrease in the near-surface soil temperatures.

Line 350: Any ideas why it decreased only at MH-S?

**Response:** There are two main reasons for this. First, at the MH-S site, ALT had reached 4.0-5.5 m from 2017 to 2020, and depth of influence of air temperature on ground temperature did not reach this depth. It can be seen from the AL-S site (Figure 8b) that ALT basically does not fluctuate. Secondly, at MH-S site, it had been 30-33 years after fire, vegetation had recovered well, and both ALT and soil temperature should be in the recovery stage, so ALT is in a downward trend. Other sites, ALT had been increasing due to climate change and fires.

Figure 9: I'd like to see these results described in terms of the chronosequence and how time since fire impacts soil moisture. Soil moisture varies over short temporal and spatial distances, so it would help add depth to your one off soil moisture measurements.

**Response:** When we submitted the manuscript for the first time, we analyzed and explained terms of the chronosequence. However, according to the requirements of the editor, this is a data description article, which cannot be analyzed and explained in great detail, so we re-wrote it in the present, more brief form. However, in order for the reader to see the trend of SMC over time, we added the post-fire time to the figures.

[Figure]

Figure 9. Variations in gravimetrically-based soil moisture contents (SMC) with different fire severity at eight sites in Mangui (MG), Alongshan (AL), Gulian (GL), and Mo'he (MH) on the western flank of the northern Da Xing'anling Mountains in Northeast China in 2016. Notes: The symbol U stands for unburned, S for severely burned, and; SMC, for soil gravimetric moisture content.

Figure 10: Same as above, it would be great if these results were described in terms of the chronosequence.

**Response:** Similar to the Figure 9 (SMC), we added the post-fire time to the figure.

[Figure]

Figure 10. Variations in soil nutrients at eight sites in Mangui (MG, a to d), Alongshan (AL, e to h), Gulian (GL, i to l), and Mo'he (MH, m to p) on the western flank of the northern Da Xing'anling Mountains in Northeast China in 2016.

Notes: The symbol U stands for unburned, and S for severely burned. SOC stands for soil organic carbon; TN, for total nitrogen; TP, for total phosphorus, and; TK, for total potassium.

Line 425: Here you say SMC is decreasing, but you only have one measurement in time. How can you say it is decreasing? You haven't described the chronosequence at all in the results, so I don't think it's fair to conclude this.

**Response:** From Figure 9, SMC at severe burned sites were lower than those of unburned sites. Thus, the SMC decreased after the fire. This sentence does not mean that the "SMC is decreasing", because the preposition "by" need to follow by a gerund. Thus, the sentence is the

"*This is evidenced* **by** *increasing ground temperature, thickening active layer, decreasing SMC, and evidently changing soil nutrient contents.*" In Lines 459-460.

---

## Referee Report (RR1)

I was pleased to review the revision of this manuscript. In most cases, I feel that the authors responded to my suggestions and answered my comments. I also understand now why the authors did not include analysis of the data including the chronosequence. The points below are where I think additional information is still required.

Line 79-80: Post fire soil moisture is more complex that you've described in the introduction. I don't feel that referencing one paper is sufficient. Most of what I've seen in the literature suggests that soil moisture is typically higher at burned sites than at unburned sites as a result of reduced evapotranspiration.

Line 80-83: The addition of this sentence does not address my previous comment "A relatively large amount of organic layer remains after the fires at all of the site (minimum 20 cm). I think it's important to note this somewhere in the paper, as this minimizes post-fire changes (less active layer thickening and ground temperature increase) than if, for example, less than 5 cm remains." It's not about the vegetation regeneration, rather it's about the antecedent organic layer thickness and what remains after the fire. More context is needed.

Line 94-102: This doesn't fully address my comment. My concern is that you describe permafrost and post-fire impacts very generally throughout the introduction and rest of the manuscript, without providing context that your sites are in a particular type of environment. It affects your results and conclusions. You can't make broad conclusions for all types of environments based on a certain subset of sites. I think more context needs to be provided in the introduction to put your sites in context.

Line 217: You provided more information in your response to my original comment and it would be helpful to add some to the text in this section. For your reference here is what I'm referring to from the original review:

> Line 201-203: Why were these thresholds chosen?
>
> Response: This is the common method of international fire severity division, and it is also a standard means of division. According to the Cocke et al., (2005) and Roy et al., (2006), the dNBR optimality values for these average changes are 0.241 for grass and 0.57 for shrub. Therefore, these values are selected as threshold values through the classification of fire severity by vegetation burn status and the comparison with dNBR (Key and Benson, 2006; Escuin et al., 2008).

Line 292: COVID-19 epidemic should be "pandemic".

Line 459: The addition of this text does not address my original comment fully. The original comment was "Here you say SMC is decreasing, but you only have one measurement in time. How can you say it is decreasing? You haven't described the chronosequence at all in the results, so I don't think it's fair to conclude this." I realize now why you did not describe the chronosequence (editor's request), but it still makes understanding the conclusion difficult for the reader.

---

## Author Response (AR2)

**Response Letter of X Li's to reviewer' comments on MS No. essd-2024-187**

I was pleased to review the revision of this manuscript. In most cases, I feel that the authors responded to my suggestions and answered my comments. I also understand now why the authors did not include analysis of the data including the chronosequence. The points below are where I think additional information is still required.

**Response:** Thank you for all your efforts for and time spent on our manuscript.

These suggestions are very helpful for improving the quality of our manuscript. I'm sorry that I didn't dispel all your concerns in the first revision. On the basis of your suggestions, I made some modifications to this manuscript as advised. I hope this revision will address your concerns.

Line 79-80: Post fire soil moisture is more complex that you've described in the introduction. I don't feel that referencing one paper is sufficient. Most of what I've seen in the literature suggests that soil moisture is typically higher at burned sites than at unburned sites as a result of reduced evapotranspiration.

**Response:** Agreed and done. Forest fire can increase or decrease soil moisture content. In general, the near-surface (<30 cm) soil moisture contents increased in the short term after fire, while for deep (>1 m) or in the long term, soil moisture contents decreased. Thus, we changed to the: in Lines 84-95,

"*Forest fires also can cause significant changes in soil moisture contents, which in turn affects ground thermal regimes (Nossov et al., 2013). Due to the fire-induced thaw of permafrost, the charred moss layers with lowered infiltration rates, lower transpiration rate and reduced evapotranspiration in severely burned areas, surface soil moisture contents (generally less than 30 cm in depth) at burned sites were significantly higher than those at unburned sites (Kopp et al., 2014; Potter and Hugny, 2020; Yoshikawa et al., 2003). However, affected by soil texture, permafrost thaw after fire can also lead to a decrease in soil moisture contents (Li et al., 2022b; Nossov et al., 2013). In summary, in a short term, forest fires will decrease rates of transpiration, raising soil moisture contents; in a long-term (more than a decade), the increased ALT and recovery of vegetation will reduce soil moisture content at burned sites as compared to that at unburned sites (Yoshikawa et al., 2003).*"

Line 80-83: The addition of this sentence does not address my previous comment "A relatively large amount of organic layer remains after the fires at all of the site (minimum 20 cm). I think it's important to note this somewhere in the paper, as this minimizes post-fire changes (less active layer thickening and ground temperature increase) than if, for example, less than 5 cm remains." It's not about the vegetation regeneration, rather it's about the antecedent organic layer thickness and what remains after the fire. More context is needed.

**Response:** Agreed and done. The postfire thickness of the soil organic layer and its impact on soil thermal conductivity was the most important factor for determining postfire soil temperatures and thaw depth. With the restoration of vegetation, the organic layer accumulates again and the thickness of residual organic layer after fire

has a significant influence on the permafrost. Therefore, according to your suggestion, the relevant content of organic layer thickness was mentioned in the ***Introduction*** and the ***2.1 Study area descriptions and monitoring networks***, and these sentences were added:

*"In Interior Alaska, organic layer thickness decreased from 21 to 4 cm after fire, resulting in thaw depth increasing from 72 to 152 cm, mean annual surface temperature rising from −0.6 to +2.1℃ and mean annual deep temperature going up from −1.7 to +0.4℃ (Nossov et al., 2013)."* In Lines 72-76.

*"At severe burned sites in AL, GL, and MH, measurements of organic matter thickness were taken 7, 14, and 29 years after fires, so it was possible that the organic layer thickness exceeded 20 cm due to the re-accumulation of organic matter. At severe burned site in MG, the organic matter residue after combustion was in a fluffy state with the thickness of 20 cm. When the re-accumulation or residual organic matter exceeded 20 cm, this would slow the rate of active layer thickening and soil temperature increase after fires, as well as the permafrost would gradually recover with the re-accumulation of organic layer."* In Lines 221-228.

Line 94-102: This doesn't fully address my comment. My concern is that you describe permafrost and post-fire impacts very generally throughout the introduction and rest of the manuscript, without providing context that your sites are in a particular type of environment. It affects your results and conclusions. You can't make broad conclusions for all types of environments based on a certain subset of sites. I think more context needs to be provided in the introduction to put your sites in context.

**Response:** The descriptions of all studies in this part are in the boreal forest and permafrost region, and the purpose of this example (a certain subset of sites) is to show the significant changes after fire. Our study area is also located in boreal forest and permafrost region. Therefore, we have made the following revisions, and hope to take care of your concerns. In Lines 99-102, 111-123, 124-126, 175-177.

*"It contains 1100–1500 Pg carbon in boreal permafrost regions (1 Pg=$10^{15}$ g), approximately twice of the carbon pool in the atmosphere (Hugelius et al., 2014), accounting for nearly half of the global belowground organic carbon pool (O'Donnell et al., 2011a)."*

*"Therefore, in the boreal permafrost region, wildfire exacerbate rates of permafrost thaw and alter soil organic carbon dynamics in both organic and mineral soils. In addition to soil organic carbon, forest fires potentially also reduce soil nitrogen contents, inducing shifts in nutrient cycling in the boreal forest and permafrost regions (Certini, 2005; Knicker, 2007; Kolka et al., 2017). However, there are inconsistent reports on the effects of forest fire on soil phosphorus and potassium. Some studies show a significant post-fire reduction in phosphorus and potassium while other studies indicate an evident increase after light burns, but a reduction after severe burns, and nearly unchanged stocks of potassium and phosphorus (Gu et al., 2010; Neff et al., 2005; Zhao et al., 1994). As a result, wildfires in boreal permafrost regions had been*

*considered to trigger strong positive feedbacks on climate warming via massive emissions of biogenic major greenhouse gases (Koven et al., 2015; Ramm et al., 2023)."*

*"Located on the southern margin of Eastern Asian boreal forests and permafrost regions, the Da Xing'anling (Hinggan) Mountains in Northeast China are prone to frequent and massive wildfires."*

*"A permafrost monitoring network has been established in four burned areas in the northern Da Xing'anling Mountains in Northeast China in boreal forest and discontinuous permafrost regions (Figure 1)."*

Line 217: You provided more information in your response to my original comment and it would be helpful to add some to the text in this section. For your reference here is what I'm referring to from the original review:

> Line 201-203: Why were these thresholds chosen?
> Response: This is the common method of international fire severity division, and it is also a standard means of division. According to the Cocke et al., (2005) and Roy et al., (2006), the dNBR optimality values for these average changes are 0.241 for grass and 0.57 for shrub. Therefore, these values are selected as threshold values through the classification of fire severity by vegetation burn status and the comparison with dNBR (Key and Benson, 2006; Escuin et al., 2008).

**Response:** Agreed and done. Added this sentence in the text. In Lines 254-257.

*"According to the Cocke et al. (2005) and Roy et al. (2006), the dNBR optimality values for these average changes are 0.241 for grass and 0.57 for shrub. Therefore, through vegetation burn status and the comparison with dNBR values (Key and Benson, 2006; Escuin et al., 2008), fire severity is thus divided into four categories"*

Line 292: COVID-19 epidemic should be "pandemic".
**Response:** Agreed and done. Changed the "epidemic" to the "pandemic". In Line

Line 459: The addition of this text does not address my original comment fully. The original comment was "Here you say SMC is decreasing, but you only have one measurement in time. How can you say it is decreasing? You haven't described the chronosequence at all in the results, so I don't think it's fair to conclude this." I realize now why you did not describe the chronosequence (editor's request), but it still makes understanding the conclusion difficult for the reader.

**Response:** Agreed and done. From Figure 9, SMC at the burned site is significantly lower than that of the unburned site. To be more convincing about the results, we have modified the sentence to the *"This is evidenced by rising ground temperature, thickening active layer, and evidently changing SMC and soil nutrient contents."* In Lines 490-491

---

## Author Response (AR3)

**Response Letter of X Li's to editor' comments on MS No. ESSD-2024-187**

Please carefully consider some final comments for the final version of your paper:
Lines 22-23. '… occurrences of wildfires have been increasingly more frequent in boreal and arctic forests during the last few decades'. Do the authors mean: '… occurrences of wildfires have been increasingly more frequent in boreal forests and arctic tundra during the last few decades'?
**Response:** Yes, it is our meaning.

Thus, after re-writing, it is expressed as follows in Lines 22-23:
Under a warming climate, occurrences of wildfires have been increasingly more frequent in boreal forests and arctic tundra during the last few decades.

Lines 99-103. 'It contains 1100–1500 Pg carbon in boreal permafrost regions (1 Pg=1015 g), approximately twice of the carbon pool in the atmosphere (Hugelius et al., 2014), accounting for nearly half of the global belowground organic carbon pool (O'Donnell et al., 2011a).' Please rephrase as something like: 'Arctic-boreal permafrost soils contain between 1100-1500 Pg carbon,…'.
**Response:** Agreed and done. Changed to the
In Lines 99-102.
"Arctic-boreal permafrost soils contain between 1100-1500 Pg (1 Pg=1015 g) carbon, approximately twice of the carbon pool in the atmosphere (Hugelius et al., 2014), and accounting for nearly half of the global belowground organic carbon pool (O'Donnell et al., 2011a)."

Lines 166-167. '…and; in Section 5, major conclusions and prospects.' Please add 'are given' or something similar at the end of the sentence.
**Response:** Agreed and done. Changed to the
In Lines 166-167
"…and; in Section 5, major conclusions and prospects are given."

Lines 191-192. 'The light blue areas in Figure 1a is the permafrost region.' Should be 'The light blue areas in Figure 1a are the permafrost region.'
**Response:** Agreed and done. Changed to the
In Lines 191-192
"The light blue areas in Figure 1a are the permafrost region."

Lines 254-255. 'According to the Cocke et al. (2005) and Roy et al. (2006), the dNBR optimality values for these average changes are 0.241 for grass and 0.57 for shrub.' dNBR optimality should not be confused with dNBR values. The dNBR optimality following Roy et al. (2006) offers a peformance metric for the dNBR index. I suggest the authors rewrite this section and remove ambiguity between the terms dNBR values and dNBR optimalilty.
**Response:** Agreed and done. Changed to the
In Lines 254-256.
"According to the Cocke et al. (2005) and Roy et al. (2006), the dNBR values of 0.241 and 0.57 are the critical values for the division between lightly and moderately burned,

and moderately and severely burned."

Lines 328-329. '… a total of 28,890 pieces of data were collected'. Suggested rephrasing: "… a total of 28,890 data records were collected'.
**Response:** Agreed and done. Changed to the
In Lines 328-329
"…, a total of 28,890 data records were collected, …"